# Faithfulness Under the Distribution: A New Look at Attribution Evaluation

Zhiyu Zhu[1], Zhibo Jin[1], Jiayu Zhang[2], Bartlomiej Sobieski[3], Przemyslaw Biecek[3]

Fang Chen[1], Jianlong Zhou[1*]

## Abstract

Evaluating the faithfulness of attribution methods remains an open challenge. Standard metrics such as Insertion and Deletion Scores rely on heuristic input perturbations (e.g., zeroing pixels), which often push samples out of the data distribution (OOD). This can distort model behavior and lead to unreliable evaluations. We propose FUD, a novel evaluation framework that reconstructs masked regions using score-based diffusion models to produce in-distribution, semantically coherent inputs. This distribution-aware approach avoids the common pitfalls of existing Attribution Evaluation Methods (AEMs) and yields assessments that more accurately reflect attribution faithfulness. Experiments across models show that FUD produces significantly different—and more reliable—judgments than prior approaches. Our implementation is available at: https://github.com/LMBTough/FUD.

## 1 Introduction

With the widespread adoption of deep learning techniques in critical domains such as medical diagnosis (Bakator & Radosav, 2018), financial risk control (Mashrur et al., 2020), autonomous driving (Grigorescu et al., 2020), and security surveillance (Yu et al., 2021), the issue of model interpretability has attracted increasing attention. Although deep neural networks have achieved breakthrough performance across a range of tasks, their complex internal mechanisms are often regarded as a "black box," making it difficult to understand the rationale behind specific predictions. In safety-critical or high-risk applications, a lack of reasonable explanation for model decisions may lead to severe consequences (Van der Velden et al., 2022). Therefore, enhancing model interpretability not only helps to foster user trust, but also facilitates error analysis, model debugging, and even improves robustness and generalization performance.

Among the various explainability techniques, attribution methods have emerged as a crucial approach for interpreting complex models by mapping prediction outcomes back to the input space (such as image pixels or feature dimensions) to identify key regions or factors that the model focuses on. These methods are widely employed in tasks like image classification (Rao et al., 2022) and sentiment analysis (Pan et al., 2024), and have demonstrated significant value, particularly in scenarios that demand high levels of security in model decision-making.

However, attribution methods themselves are not always reliable or consistent, as different techniques may produce significantly divergent explanations for the same model prediction. Therefore, how to objectively assess the quality of attribution results has emerged as a core challenge in current research. Faithfulness (Petsiuk et al., 2018; Yeh et al., 2019) is widely regarded as a key criterion for evaluating the effectiveness of attribution methods. Such criterion reflects the consistency between the attribution results and the actual decision-making basis of the model: a highly faithful attribution method indicates that the highlighted high-contribution regions genuinely play a critical role in the model's prediction. In other words, when these regions are removed or perturbed, the model's output changes significantly.

Building on this insight, the research community has proposed various quantitative metrics to assess faithfulness. Among them, the most representative and widely adopted are the Insertion & Deletion

---

*Corresponding author: jianlong.zhou@uts.edu.au; [1]University of Technology Sydney; [2]Suzhou University of Technology; [3]Warsaw University of Technology.

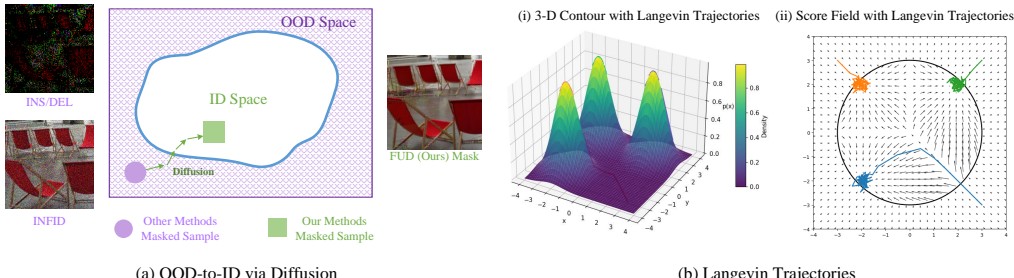

(a) OOD-to-ID via Diffusion                                      (b) Langevin Trajectories

Figure 1: **Overview of the FUD framework.** (a) The left panel illustrates the core motivation of FUD: compared to conventional attribution evaluation metrics such as *INS/DEL* and *INFID*, which generate perturbed inputs drifting into the OOD space (purple area), FUD ensures that evaluation samples remain within the ID manifold (green area) by reconstructing masked regions via a score-based diffusion model. (b) The right panel shows how FUD leverages Langevin dynamics to guide inputs back toward the data manifold. Subfigure (i) visualizes Langevin trajectories in a toy 3D density; subfigure (ii) shows how the learned score field $\nabla_x \log p_\theta(x)$ steers samples from OOD to ID regions. The derivation of this score function is discussed in Section 3.3.

Scores (Petsiuk et al., 2018) and Infidelity (Yeh et al., 2019). The Insertion & Deletion Score is a pair of complementary evaluation metrics, which start from the "substitutability of contribution areas" and simulate the way humans understand the model's dependent areas. Specifically, the Deletion Score progressively removes the highest-scoring regions from the original input—i.e., the pixels or features with the largest attribution values—and records the corresponding changes in the model's output at each step. In contrast, the Insertion Score starts from a blank image (such as an all-zero input or a blurred background) and gradually inserts the highest-scoring regions from the attribution map in order of decreasing attribution value.

Infidelity is another faithfulness metric based on expected values, focusing on the explanatory power of attribution scores with respect to prediction changes. It is defined as the mean squared error between the attribution values and the actual changes in the model's predictions under a set of input perturbations, such as additive noise. Although existing Attribution Evaluation Methods (AEMs) provide powerful tools for quantitatively analyzing attribution methods, they also suffer from notable limitations. These methods fundamentally rely on the core operation of feature removal or modification, a process that inherently introduces two key heuristic mistakes, **introducing additional information** and **being affected by the instability of the out-of-distribution space of models**. We will discuss the details later. As a result, mainstream attribution methods often fail to faithfully reflect the true decision-making basis of the model, leading to explanations that lack faithfulness. To address the shortcomings of current AEMs in accurately capturing the faithfulness of attribution algorithms, inspired by Score-based Generative Modeling (SGM), we propose a new evaluation method called FUD. FUD is capable of constructing attribution evaluation samples that remain within the data distribution while preserving the essential information required for faithful evaluation. The core idea of FUD lies in its ability to leverage the score function learned from the true data distribution to pull out-of-distribution samples back into the data manifold, while simultaneously preserving the critical information needed for faithful attribution evaluation. Building on this idea, we present a detailed derivation of the distributional formulation underlying FUD. It is worth emphasizing that FUD can be readily applied by incorporating the target model into an existing score function, without the need for any additional training. An overview of the proposed pipeline is shown in Fig. 1.

**Contributions.** (1) we first identify that widely used Attribution Evaluation Methods (AEMs) systematically exhibit significant heuristic issues, posing substantial risks to explainable AI, and we analyse these issues from a distributional perspective using explicit OOD detection and image–quality metrics; (2) we propose and formally derive the FUD evaluation method, explicitly designed to avoid these heuristic pitfalls, and validate its effectiveness through extensive experiments across diverse use cases; and (3) we release FUD as an open-source toolkit to promote transparency, reproducibility, and community engagement.

## 2 Related Work

Attribution methods aim to explain model predictions by identifying input regions most responsible for the output, with approaches such as Integrated Gradients (IG) (Sundararajan et al., 2017), Guided IG (Kapishnikov et al., 2021), Boundary-based IG (Wang et al., 2021), and adversarial-path methods including AGI (Pan et al., 2021) and subsequent adversarial attribution variants (Zhu et al., 2024b;a;c). Evaluating attribution faithfulness typically relies on perturbation-based metrics such as Insertion & Deletion Scores (Petsiuk et al., 2018), Infidelity (Yeh et al., 2019), Sensitivity-$n$ (Ancona et al., 2017), and optimized-mask approaches (Fong & Vedaldi, 2017; Fong et al., 2019), all of which introduce distribution shifts to varying degrees. In addition, generative in-filling approaches (Chang et al., 2019; Agarwal & Nguyen, 2020) use learned generative models to replace removed pixels with realistic content; however, they do not incorporate classifier–input gradients and may inadvertently introduce class-supporting evidence instead of faithfully removing it. These methods collectively highlight the trade-off between interpretability and distributional consistency, motivating more robust evaluation frameworks. A detailed and extended discussion of attribution methods and evaluation metrics is provided in Appendix A.

## 3 Method

### 3.1 Problem definition

In the Research of interpretability for deep learning models, attribution methods aim to measure the contribution of input features to the model's decision. Given a trained predictive model $f(x)$, where $x \in \mathbb{R}^n$ represents an $n$-dimensional input sample, the model's output is a $c$-dimensional vector $f(x) \in \mathbb{R}^c$, corresponding to the probability distribution or prediction scores over $c$ classes. Attribution methods attempt to generate an interpretable representation $A(x) \in \mathbb{R}^n$ for the input sample $x$, indicating the importance of each input feature or feature region for the model output (usually for a specific class). For the $i$-th feature in the image, a larger value of $A(x)_i$ represents a greater contribution of that feature to the model's decision. In the Appendix A, we also provide a detailed related work of the current state-of-the-art attribution methods and commonly used AEMs.

**Model-centric out-of-distribution (OOD) definition.** Throughout this paper, OOD is defined with respect to the original model under evaluation. That is, a perturbed sample is considered in-distribution if it lies on (or near) the model's learned data manifold, regardless of whether it appears visually "natural" to humans. This model-centric view is consistent with our goal—assessing attribution faithfulness for the original model—rather than for a re-trained or adaptively fine-tuned model. See Appx. F for objective OOD-detection evidence supporting this distinction.

### 3.2 The heuristic mistakes of current AEMs

Most of the currently designed AEMs (Petsiuk et al., 2018; Yeh et al., 2019; Ancona et al., 2017) are heuristic in nature. Although they are often supported by some mathematical interpretations and theoretical foundations (Yeh et al., 2019), they are essentially based on intuition-driven designs. Of course, we do not consider heuristic or intuition-driven evaluation method design to be a wrong choice. Just as using the probability output of a classification model as a measure of confidence is widely accepted—it effectively reflects the relative likelihoods of different class decisions made by the model. Since AEMs require an understanding of the model's internal mechanisms, and such understanding still largely relies on heuristic reasoning and empirical analysis, we inevitably make reasonable assumptions about these internal processes during interpretation. However, because these assumptions are themselves heuristic, it is difficult to design AEMs based on first principles. But precisely because of this, we need to approach the potential heuristic mistakes with greater rigor.

We find that mainstream AEMs commonly involve feature insertion and deletion during their design. For example, the insertion score is calculated by progressively inserting features based on their attribution-estimated contribution values in descending order, and observing the increase in the confidence score of the target class. A higher insertion score indicates that the attribution method has successfully identified important, high-contribution features. Similarly, the deletion score involves removing features in descending order of their estimated contribution values, where a lower score

indicates a better attribution method. During this process, a corresponding mask $M \in {0,1}^n$ is generated, where the ratio of 0 and 1 depends on the number of features we want to retain. The dimensions with a value of 1 indicate the features that are intended to be kept. Furthermore, we directly construct an evaluation sample $\tilde{x} = M \odot x + (1-M) \odot \mathbf{0}$, which lies outside the distribution that the model is responsible for. Intuitively, this is an appropriate evaluation method that can directly reflect the impact of feature importance.

However, in reality, this process is based on a fundamentally flawed assumption—**that removing a feature means setting it to zero** (the insertion score can be seen as the dual counterpart of the deletion score, which removes unimportant features). In images, zero often represents specific color information, such as black, so **setting values to zero does not equate to removing features**; instead, it may introduce new, semantically meaningful information into the image. For a straightforward example: in a classification task distinguishing **black cats** from **white cats**, the black regions are important features for recognizing the "black cat" class. If we set parts of the image to zero (i.e., turn them black), this does not actually remove the black information—in fact, it may even strengthen the representation of the "black cat." As a result, the model's confidence output might not decrease and could even increase, which directly contradicts our expectation that the model should fail when important features are removed.

Aside from the issue of introducing additional information, current AEMs also suffer from another obvious heuristic mistake: **it is difficult to ensure that samples with removed features remain valid and realistic samples**. We assume that we have removed 50% of the important features, resulting in images containing half blacked-out regions. However, such samples would never appear in reality. When training the model, we expect it to learn the distribution $P_{x \sim P(x)}(y|x)$,, where $P(x)$ represents the true data distribution, and $y$ denotes the class label information. In other words, since the true data distribution occupies only a very small portion of the high-dimensional space, the model only needs to—and can only—be responsible for the distribution of the training data, while a large amount of uncertainty exists outside this distribution. Using model behavior outside the data distribution to evaluate model behavior within the distribution is a highly counterintuitive approach.

In Section 4.3.1 and Section 4.3.2, we analyzed the intermediate images generated by current attribution evaluation methods using OOD detection techniques and image quality evaluation metrics, and found that these samples not only lie outside the true data distribution but also appear highly unrealistic. Since the samples with removed information often lie far from the true distribution $P(x)$, the shift in the model's output probability distribution during this process may stem not only from biases in the class decision itself but also from discrepancies between the sample and the true distribution $P(x)$. This issue cannot be distinguished during the attribution evaluation process, which undermines the credibility of AEMs. Moreover, because these evaluation results lie in an unstable OOD space, manifesting as very unsmooth insertion and deletion curves, including abnormal phenomena such as confidence increasing when important features are deleted.

Besides Insertion& Deletion Scores, evaluation methods such as Infidelity and Sensitivity-$n$, although attempting to circumvent the explicit insertion/deletion issues by focusing on consistency or sensitivity, still rely on intervening in the input features during their operations. Consequently, they inevitably introduce intermediate images that deviate from the original distribution, resulting in similar heuristic mistakes in the evaluation outcomes. The issues with the Sensitivity-$n$ metric are similar. Although this metric observes changes in model predictions by randomly occluding a subset of important features selected based on attribution rankings, it fundamentally relies on the assumption that "occlusion is equivalent to feature deletion." . Meanwhile, these occlusion operations may still introduce unnatural semantic cues or create abnormal structures within the images, thereby affecting the stability of the model's output. In summary, when designing AEMs, we must rigorously avoid these heuristic mistakes.

## 3.3 FAITHFULNESS UNDER THE DISTRIBUTION (FUD)

### 3.3.1 INTUITION

The core idea of FUD is simple: given an attribution map, we construct perturbed samples that (i) remain on the data manifold and (ii) preserve all visible features that the attribution method identifies as important, without introducing new evidence that supports the predicted class $y$. Conceptually, this

corresponds to a diffusion-style inpainting process guided by a hard mask and by a "no new evidence for class $y$" bias, so that the evaluation reflects only the contribution of the preserved features.

For completeness, we note that such perturbations are expected to satisfy several desirable properties, including in-distribution realism, exact preservation of the retained features, absence of hallucinated class evidence, and perceptual coherence. These desiderata motivate the FUD update rule, and are discussed in more detail in Appendix B.

### 3.3.2 DERIVATION

We now summarise the FUD evaluation algorithm and formalise the above intuition. The derivation addresses two questions: how to keep the perturbed samples within the data distribution, and how to preserve the information we want to evaluate.

**Within the distribution**  Inspired by the SGM algorithm (Song et al., 2020), we assume that the true data distribution is $P(x)$. We denote by $x^t$ an intermediate sample at iteration $t \in \{1, 2, \ldots, T\}$, where $t$ counts the remaining update steps and each update transforms $x^t$ into $x^{t-1}$. We initialise

$$x^T = M \odot x + (1 - M) \odot \epsilon, \qquad \epsilon \sim \mathcal{N}(0, I), \tag{1}$$

so that the unmasked pixels follow the original image $x$, while the masked pixels are replaced by noise; in general, such an $x^t$ lies off the data manifold.

If we could obtain the gradient $\nabla_{x^t} P(x^t)$ of the true distribution at the current position, we could use gradient ascent to update $x^t$ and guide it towards regions of higher density. In practice, gradients in the input space are very sparse and the optimisation must satisfy normalisation constraints, so we instead work with the score function and update according to

$$x^{t-1} = x^t + c \nabla_{x^t} \log P(x^t) + \sqrt{2c}\, \epsilon, \qquad \epsilon \sim \mathcal{N}(0, I), \tag{2}$$

where $c$ denotes the Langevin step size and the noise term follows Song et al. (2020). We approximate the true score $\nabla_{x^t} \log P(x^t)$ using a learned score function $s_\theta(x^t)$, trained with the standard SGM objective

$$\theta^* = \arg\min_\theta \sum_{t=1}^{T} \lambda(t) \, \mathbb{E}_{P_{\sigma^t}(x^t)} \left[ \left\| s_\theta(x^t) - \nabla_{x^t} \log P_{\sigma^t}(x^t) \right\|_2^2 \right], \tag{3}$$

where $\lambda(t)$ and $\sigma^t$ are pre-defined hyperparameters. The score $s_\theta$ can also be learned via diffusion models such as DDPM (Ho et al., 2020). Since this paper does not focus on the training details, we refer the reader to SGM (Song et al., 2020); here it is sufficient to note that, once we obtain the gradient information of the distribution, we can update samples using gradient steps to bring them back within the distribution.

**Preservation of the evaluation information**  During the evaluation of attribution methods, we progressively remove or modify features that the attribution method deems unimportant, thereby preserving the features that are crucial to the model's decision-making. FUD aims to generate evaluation samples that both stay within the data distribution and preserve precisely these features. To achieve this, we require that the features to be evaluated are kept intact, and that no newly generated features bring the sample closer to a different class distribution. Otherwise, it becomes impossible to disentangle the effect of the original preserved features from that of newly created, class-supporting evidence.

To enforce this constraint, we introduce a hypothetical distribution $\tilde{P}(y \mid x^t)$ whose gradient is opposite to that of $P(y \mid x^t)$,

$$\nabla_{x^t} \tilde{P}(y \mid x^t) = -\nabla_{x^t} P(y \mid x^t), \tag{4}$$

and we denote this "no new class evidence" bias as an event $z$. The target distribution that FUD aims to sample from is then

$$
\begin{aligned}
P(x^t \mid z, \tilde{x}, M) &= \frac{P(z, \tilde{x} \mid x^t, M)\, P(x^t \mid M)}{P(z, \tilde{x} \mid M)} \\
&= \frac{P(z \mid x^t)\, P(\tilde{x} \mid x^t, M)\, P(x^t)}{P(z, \tilde{x} \mid M)} \; \propto \; P(x^t)\, P(z \mid x^t)\, P(\tilde{x} \mid x^t, M),
\end{aligned}
\tag{5}
$$

where $P(z, \tilde{x} \mid M)$ is a normalising constant and $P(x^t \mid M) = P(x^t)$ if the prior is independent of $M$. Taking gradients yields

$$\nabla_{x^t} \log P(x^t \mid z, \tilde{x}, M) = \nabla_{x^t} \log P(x^t) - \nabla_{x^t} \log P(y \mid x^t) + \nabla_{x^t} \log P(\tilde{x} \mid x^t, M), \quad (6)$$

with all gradients taken with respect to $x^t$.

In this expression, the first term $\nabla_{x^t} \log P(x^t)$ is the image prior $P(x)$ (not a label vector) and is provided by the learned score network $s_\theta(x^t)$. The second term $\nabla_{x^t} \log P(y \mid x^t)$ is the classifier's input gradient with respect to $x^t$, supplied by the model whose behaviour we evaluate; in later steps, we only activate this term after $x^t$ has moved sufficiently close to the data manifold. The third term $\nabla_{x^t} \log P(\tilde{x} \mid x^t, M)$ enforces consistency on the unmasked pixels. Choosing

$$P(\tilde{x} \mid x^t, M) = \prod_{M_i = 1} \delta(x_i^t - \tilde{x}_i) \quad (7)$$

ensures that these coordinates remain fixed, effectively combining the score-model prior, the classifier-gradient correction, and the hard masking constraint into a single update rule in Eq. (2). In implementation, we directly replace the entries of $x^t$ with $\tilde{x}$ whenever $M_i = 1$; further properties are provided in Appendix C.

The construction of $\nabla_{x^t} \log P(y \mid x^t)$ can be seen as taking the gradient of the input with respect to the negative cross-entropy loss. However, the model behaviour is reliable only for samples that lie within the data distribution. The initial sample $x^T$ is far from the manifold, and the corresponding $\nabla_{x^t} \log P(y \mid x^t)$ is therefore not meaningful. In SGM-like approaches, similar gradients require additional training that sacrifices classification performance in order to obtain useful gradients on noisy samples, which is incompatible with evaluating arbitrary pretrained models. To avoid this, we initially ignore the $z$-term and use

$$P(x^t \mid \tilde{x}, M) = \frac{P(\tilde{x} \mid x^t, M) \, P(x^t)}{P(\tilde{x} \mid M)} \; \propto \; P(x^t) \, P(\tilde{x} \mid x^t, M) \quad (8)$$

instead of $P(x^t \mid z, \tilde{x}, M)$, using only the score prior and mask constraint to move samples closer to the manifold. In Appendix D, we show empirically, using the score function to generate evaluation samples, that after updating $P(x^t \mid \tilde{x}, M)$ for about $5\%$ of the remaining sampling steps, the samples begin to enter the in-distribution region. At that point, features close to the original class distribution have not yet been newly generated, so we switch to the original target distribution $P(x^t \mid z, \tilde{x}, M)$ and continue sampling to obtain high-quality evaluation samples.

Finally, we summarise how FUD is used as an evaluation metric. FUD generates attribution–evaluation samples that stay within the distribution while preserving the features that need to be evaluated. We follow a deletion-style protocol: we progressively remove features deemed unimportant by the attribution method, use FUD (with fixed hyperparameters) to generate the corresponding evaluation samples at each removal level, and track the model's confidence on these samples. If an attribution algorithm can accurately identify important features, then evaluation samples retaining the same proportion of features will exhibit higher confidence. We only consider this "retain-important-features" direction, rather than defining separate insertion and deletion scores, because evaluating samples where only unimportant features are kept is often not informative (for example, keeping a few background patches of grass in a black-cat vs. white-cat task). From an optimisation perspective, the presence of the $-\nabla_{x^t} \log P(y \mid x^t)$ term also means that explicitly evaluating "unimportant" features would tend to amplify adversarial effects and destabilise the metric, making it difficult to distinguish truly unimportant features from artefacts introduced by this term. The pseudocode of our FUD algorithm is provided in Appendix E.

## 4 EXPERIMENT

### 4.1 EXPERIMENTAL SETUP

**Models & Data.** To demonstrate that the proposed evaluation scheme generalises across network families, we test one convolutional and one transformer backbone: RESNET-50 (He et al., 2016)—the canonical ImageNet convolutional network—and VIT-B/16 (Dosovitskiy, 2020), a vision transformer of comparable capacity. Both checkpoints are the publicly released ImageNet-1k weights and are

Table 1: Mainstream attribution evaluation metrics compared with FUD.

| Metric | Description |
| --- | --- |
| Insertion & Deletion (INS/DEL) | Measures change in model output after iteratively inserting / deleting high-score regions. |
| Infidelity (INFID) | Expected squared error between attribution-weighted perturbations and output change. |
| Sensitivity-$N$ (Sen-N) | Correlation between output change and random masking of the top-$N$ salient features. |

kept *frozen* during every attribution run, thereby eliminating confounding factors that fine-tuning could introduce. Following Pan et al. (2021) and Long et al. (2022), we draw 1,000 validation images uniformly at random from ImageNet (Deng et al., 2009). No further curation is performed, so the subset retains the long-tailed distribution of object categories and recording conditions; each attribution method is therefore evaluated on exactly the same, unbiased sample.

**Attribution Baselines.** We benchmark eleven representative explainability techniques that collectively cover gradient-, perturbation-, and attack-based families: FIG (Hesse et al., 2021), GIG (Kapishnikov et al., 2021), IG (Sundararajan et al., 2017), BIG (Wang et al., 2021), SM (Simonyan et al., 2013), MFABA (Zhu et al., 2024c), ATTEXPLORE (Zhu et al., 2024b), ISA (Zhu et al., 2024a), EG (Erion et al., 2021), AGI (Pan et al., 2021), and LA (Zhu et al., 2024d). All algorithms are executed with the hyper-parameters recommended by their authors.

**Competing Evaluation Criteria.** To test whether the **proposed metric (FUD)** in Section 3 yields a more faithful signal of explanation quality, we compare it against three widely used quantitative criteria, summarised in Table 1.

**Implementation Details.** All experiments run on two NVIDIA L40S GPUs (48 GB) with PyTorch 2.4.1. Mixed-precision inference (FP16) is enabled wherever supported, yielding a $\sim 1.7\times$ speed-up without compromising numerical stability. Following standard practice in generative-model evaluation, we adopt deterministic image-quality metrics (PSNR, SSIM, MS-SSIM, FSIM, GMSD, HaarPSI, VSI) that return fixed scores for each image pair, computed via the PyTorch Image Quality library with default configurations. We set the random seed to 3407 for reproducibility, following Picard (2021). Results are reported as averages over all test samples, consistent with diffusion-model evaluation protocols (Ho et al., 2020).

**FUD Diffusion Configuration.** FUD employs an unconditional diffusion generator `256x256_diffusion_uncond.pt`. The underlying U-Net has 256 base channels, two residual blocks per resolution, and multi-head self-attention at 32, 16, and 8 pixels (head dim. 64). All residual blocks include up/down sampling with scale-shift normalisation. We adopt FP16 arithmetic and a 1 000-step linear noise schedule while learning both mean and variance (`learn_sigma=True`). Although the generator is class-agnostic, we steer the reverse process with a pretrained classifier (`256x256_classifier.pt`); a guidance scale of 4.0 plus classifier-free guidance weight 2.0 balances diversity and fidelity. Each of the 1 000 ImageNet images is explained at $256\times256$ resolution with batch size 2.

## 4.2 COMPUTATIONAL CONSIDERATIONS

While FUD provides more reliable attribution evaluations by ensuring in-distribution samples, it introduces additional computational overhead compared to standard deletion/insertion methods. Generating diffusion-based samples for each masking step requires approximately a few seconds per image on modern GPUs, whereas traditional zero-masking is virtually instantaneous. This trade-off between evaluation fidelity and computational efficiency is inherent to our approach. In practice, the generative model needs to be trained only once per dataset and can be reused across all evaluations, amortizing the initial cost. We view this overhead as a worthwhile investment for obtaining more faithful attribution assessments. Appendix I.3 (Table 10) reports wall-clock times on ViT-B/16 for different numbers of diffusion steps.

## 4.3 EXPERIMENTAL RESULTS

We conduct extensive experiments to validate the proposed FUD evaluation scheme against existing attribution evaluation metrics. Unless otherwise specified, all reported results are averaged over 198 runs for fairness (complete raw results are available in the supplementary repository). We compare

FUD with three widely-used evaluation methods – Insertion/Deletion (INS/DEL), Sensitivity-N (Sen-N), and Infidelity (INFID) – using two representative models (RESNET-50 and VIT-B/16) and eleven attribution methods spanning gradient- and perturbation-based explainer families. In the following, we analyze the authenticity of intermediate samples and the stability of the evaluation process, and we examine an ablation on our hard vs. soft mask constraints.

### 4.3.1 COMPARISON WITH EXISTING EVALUATION METRICS

Our first experiment demonstrates that conventional evaluation metrics produce intermediate samples that significantly deviate from the training distribution, whereas FUD generates intermediate samples that remain largely *in-distribution*. We verify this claim by employing Energy (Liu et al., 2020) as our out-of-distribution (OOD) detector to distinguish intermediate samples from genuine in-distribution (ID) data. The results in Table 2 show that intermediate inputs produced by INS/DEL, Sen-N, and INFID are easily recognized as OOD, while those produced by FUD are much harder to distinguish from normal inputs. For instance, under RESNET-50, the OOD detector achieves a high AUROC of 0.8974 on INS/DEL samples, but only 0.6863 on FUD's samples (closer to 0.5, which indicates random guessing). For completeness, additional perceptual and structural fidelity comparisons are provided in Table 9 (Appendix I.1), showing that FUD yields substantially higher PSNR/SSIM/FSIM scores than existing metrics.

Similarly, the detector's false positive rate at 95% TPR (FPR$_{95}$) jumps from 0.36 (INS/DEL) to 0.83 for FUD, and FUD yields markedly lower AUPR-In/Out values than others. In other words, FUD's transitional examples are so realistic that the detector struggles to tell them apart from ID data, whereas other methods produce "artificial" inputs with obvious OOD characteristics that are readily identified. The reason is that traditional metrics rely on naive feature removal or perturbation (e.g. replacing content with zeros or blurred backgrounds), which introduces semantic biases (such as large black or noisy regions) not seen in the training distribution. This distribution shift can spuriously alter the model's behavior, undermining the fidelity evaluation. By contrast, FUD leverages a learned score function to gradually nudge perturbed inputs back towards the data manifold while preserving the important features, thereby yielding much more authentic intermediate samples. As a qualitative illustration, we provide examples of the progressive masking process under INS/DEL versus FUD in the Appendix F; as noted in prior work (Jacovi & Goldberg, 2020), faithful explanations should be

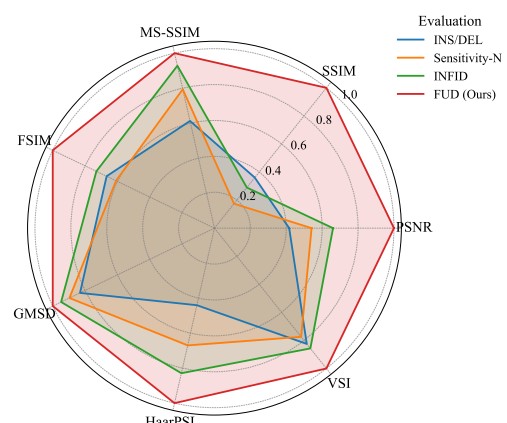

Figure 2: Radar plot of normalised image-quality metrics (GMSD inverted). FUD encloses the largest area, indicating the highest overall fidelity.

grounded in the model's representation space rather than human visual preferences, so even when extreme deletion ratios make some FUD samples appear less visually intuitive, they nevertheless remain on the model's in-distribution manifold. Under FUD, the model's confidence decays much more smoothly without abrupt jumps. Additional comparisons with recent OOD-aware attribution evaluation protocols are reported in Appendix I, further confirming FUD's advantage in maintaining in-distribution intermediate samples. A detailed analysis of the $L_2$ distance between the perturbed images and the original inputs is provided in Appendix J.

### 4.3.2 AUTHENTICITY OF TRANSITIONAL SAMPLES

While the OOD detector in Section 4.3.1 confirms *distributional* realism, we further quantify the *perceptual* and *structural* fidelity of transitional images with seven standard quality metrics; a concise summary is given in Table 6, and detailed definitions can be found in Appendix F.

**Protocol.** Eleven attribution methods are evaluated on RESNET-50 and VIT-B/16. Following standard practice, we progressively mask the top-ranked pixels from 10% to 90% (step 10%), yielding

Table 2: OOD detection performance for intermediate evaluation samples (higher values indicate easier detection as OOD). We report AUROC (higher means better OOD discrimination), $FPR_{95}$ (false positive rate at 95% true positive rate), AUPR-In, and AUPR-Out for a standard OOD detector distinguishing intermediate samples (OOD) from normal validation images (ID). **Bold** highlights the values indicating the **most ID-like (hardest to detect)** samples in each case. *Note:* In this context, AUROC, AUPR-In, and AUPR-Out values approaching 0.5 indicate maximal uncertainty in distinguishing between ID and OOD samples, thus reflecting increased in-distribution similarity and greater detection difficulty.

| | RESNET-50 | | | | VIT-B/16 | | | |
|---|---|---|---|---|---|---|---|---|
| Evaluation | AUROC ↓ | FPR$_{95}$ ↑ | AUPR-In ↓ | AUPR-Out ↓ | AUROC ↓ | FPR$_{95}$ ↑ | AUPR-In ↓ | AUPR-Out ↓ |
| INS/DEL | 0.8974 | 0.3603 | 0.8948 | 0.8893 | 0.8784 | 0.4761 | 0.8529 | 0.8894 |
| Sen-N | 0.8773 | 0.5450 | 0.8579 | 0.8864 | 0.8781 | 0.5660 | 0.8568 | 0.8734 |
| INFID | 0.7801 | 0.7720 | 0.7526 | 0.7876 | 0.8181 | 0.7390 | 0.7904 | 0.8119 |
| FUD (Ours) | **0.6863** | **0.8317** | **0.6660** | **0.6922** | **0.6450** | **0.9404** | **0.5827** | **0.6812** |

Table 3: Average image-quality scores of transitional samples produced by four evaluation metrics. Best values are **bold**.

| Evaluation | PSNR ↑ | SSIM ↑ | MS-SSIM ↑ | FSIM ↑ | GMSD ↓ | HaarPSI ↑ | VSI ↑ |
|---|---|---|---|---|---|---|---|
| INS/DEL | 10.49 | 0.27 | 0.48 | 0.58 | 0.271 | 0.292 | 0.780 |
| Sensitivity-N | 13.63 | 0.13 | 0.62 | 0.53 | 0.214 | 0.444 | 0.732 |
| INFID | 16.64 | 0.22 | 0.72 | 0.63 | 0.169 | 0.550 | 0.810 |
| **FUD (Ours)** | **25.20** | **0.75** | **0.78** | **0.86** | **0.124** | **0.663** | **0.946** |

$11 \times 2 \times 9 = 198$ runs per metric. We average each metric across all runs, obtaining Table 3, and visualise the normalised values in Figure 2.

**Discussion.** Across *all seven* metrics FUD outperforms prior evaluations by a substantial margin. Its PSNR is $+8.6$ dB higher than the next best (INFID), SSIM improves by $\sim 0.53$, and the distortion-oriented GMSD drops by $> 25\%$. Qualitatively, FUD's intermediate images preserve fine texture and colour consistency, whereas INS/DEL and Sensitivity-N introduce large black or noisy regions, and INFID yields blur artefacts. Combined with the OOD analysis, these results show that FUD produces transitional samples that are not only statistically in-distribution but also perceptually faithful, providing a solid foundation for reliable attribution evaluation.

### 4.3.3 SMOOTHNESS OF THE EVALUATION PROCESS

We quantitatively assess the *stability* and *monotonicity* of attribution evaluation under different metrics. Ideally, as more important features are removed, the model's confidence should decrease smoothly and monotonically; irregular rises or plateaus indicate unreliability. We measure smoothness with Kendall's $\tau$, which captures a sequence's monotonic trend via pairwise-order concordance. A value of $\tau = 1$ denotes perfectly monotonic decreasing confidence (each removal step strictly reduces the score), whereas values near $0$ indicate no clear trend (many out-of-order fluctuations). Table 4 reports $\tau$ for deletion sequences from eleven attribution methods evaluated with INS/DEL versus with FUD. FUD makes the process substantially more orderly and smooth: on RESNET-50, FUD gives $\tau > 0.8$ for most explainers, while INS/DEL often yields $\tau < 0.6$ (and as low as 0.21 for gradient-based methods). Notably, IG and GIG, which score very low under INS/DEL ($\tau \approx 0.22$), reach much higher monotonicity with FUD ($\tau \approx 0.69$). A similar trend appears on VIT-B/16; e.g., IG improves from $0.46$ to $0.78$ under FUD. The higher $\tau$ values indicate that confidence decreases more consistently as important features are removed, without the erratic jumps or counter-intuitive increases seen with traditional masking. This monotonic behavior suggests that FUD's in-distribution samples provide more stable, interpretable signals for attribution evaluation and produce a smoother confidence decay. We also report additional OOD–detection results for intermediate samples in Table 11 (Appendix I.2), which further confirm that FUD produces ID-like transitions that support smoother and more stable confidence decay. By keeping intermediate samples realistic, FUD ensures that each incremental removal yields a proportional, stable change in output, aligning with the ideal of a faithful attribution metric. A more detailed discussion of what FUD measures, together with a

Table 4: Smoothness of the evaluation process, measured by Kendall's $\tau$ (higher values indicate a more monotonic, smoother confidence decrease during feature removal). We compare eleven attribution methods evaluated under INS/DEL vs. under FUD. **Bold** numbers indicate the higher (smoother) value for each attribution method.

| Model | Eval. | AGI | LA | AttExp | BIG | EG | FIG | GIG | IG | ISA | MFABA | SM |
|-------|-------|-----|-----|--------|-----|-----|-----|-----|-----|-----|-------|-----|
| RESNET-50 | INS/DEL | 0.6496 | 0.5771 | 0.7944 | 0.6379 | 0.5730 | 0.2006 | 0.2128 | 0.2176 | 0.6983 | 0.6774 | 0.2833 |
| | FUD (Ours) | **0.8443** | **0.8515** | **0.8771** | **0.9129** | **0.8292** | **0.8529** | **0.6845** | **0.6905** | **0.8312** | **0.9259** | **0.6728** |
| VIT-B/16 | INS/DEL | 0.7374 | 0.5501 | 0.7407 | 0.7354 | 0.6495 | 0.3767 | 0.4523 | 0.4615 | 0.6629 | 0.7406 | 0.6015 |
| | FUD (Ours) | **0.9174** | **0.9060** | **0.9241** | **0.9046** | **0.8203** | **0.8654** | **0.7741** | **0.7803** | **0.8763** | **0.9206** | **0.7472** |

Table 5: Comparison of generated image fidelity under **hard** vs. **soft** mask constraints in FUD (using IG at 50% masking). Higher values indicate better quality for PSNR, SSIM, MS-SSIM, FSIM, HaarPSI, VSI, while lower is better for the distortion metric GMSD. **Bold** denotes the better result for each metric.

| Constraint | PSNR $\uparrow$ | SSIM $\uparrow$ | MS-SSIM $\uparrow$ | FSIM $\uparrow$ | GMSD $\downarrow$ | HaarPSI $\uparrow$ | VSI $\uparrow$ |
|------------|------|------|---------|------|------|---------|------|
| Hard (Ours) | **34.0335** | **0.947984** | **0.985258** | **0.970190** | **0.035163** | **0.921655** | **0.991347** |
| Soft | 27.6317 | 0.830200 | 0.951012 | 0.916191 | 0.081117 | 0.794670 | 0.971782 |

quantitative comparison of FUD scores across 11 representative attribution methods, is provided in Appendix G (Table 7).

### 4.3.4 EFFECT OF SOFT VS. HARD CONSTRAINTS ON IMAGE QUALITY

We ablate *soft* versus *hard* mask constraints in FUD. By default (hard), masked features are fixed to a baseline, i.e., a binary gate that fully removes selected features. As a soft alternative, inspired by Song et al. (2020) and related work, we fill masked regions with Gaussian noise centered at the original values. Formally, instead of $\prod_{i=1}^{m} \delta(x_i^t - \tilde{x}_i)$ to enforce $\tilde{x}_i = x_i^t$ for unmasked features, we sample $\tilde{x} \sim \mathcal{N}(M \odot x^t, \sigma^2 I)$ where $M$ is a binary mask (1=preserve, 0=remove), yielding the score $\nabla_{x^t} \log P(\tilde{x} \mid x^t, M) = \frac{M \odot (\tilde{x} - x^t)}{\sigma^2}$. This soft constraint adds noise in preserved regions, potentially smoothing transitions. However, it significantly *degrades* image fidelity. Pixel-level masks often induce incoherent noise, lowering quality. We evaluate fidelity under both settings by applying Integrated Gradients (IG) on 50% masked inputs and computing standard image-quality metrics. Table 5 shows that hard masking yields higher PSNR/SSIM/MS-SSIM/FSIM and lower GMSD; specifically, PSNR/SSIM of **34.03/0.948** vs. 27.63/0.830 for soft. These results confirm that the hard constraint produces more realistic, coherent transitional images, which is critical for reliable evaluation. Therefore, we adopt the hard constraint in FUD by default to maintain high image fidelity and stable performance.

## 5 CONCLUSION AND FUTURE WORK

**FUD** evaluates attribution maps in a distribution-aware way by reconstructing masked regions via a score-based diffusion process, keeping transitional samples on the data manifold. This yields more realistic, perceptually faithful, and smoother evaluation dynamics than heuristic baselines. Given compute limits (and to avoid perfectionism), we leave several optimizations for future work: (i) task-specific score functions $s_\theta(x^t)$ tailored to attribution evaluation—the current $s_\theta(x^t)$ is an unconditional guided-diffusion model trained on ImageNet and requires resolution alignment[1]; and (ii) more efficient samplers (e.g., DPM-Solver, DDIM) to mitigate the sample-generation bottleneck. At extreme deletion ratios, fills may look unnatural to humans due to limited context yet remain *model-in-distribution* under objective detectors; our evaluation adheres to this model-centric criterion. Future improvements in estimating the data–distribution score $\nabla_x \log P(x)$ may further reduce these limitations, as our framework can seamlessly incorporate stronger generative priors.

---

[1] https://github.com/openai/guided-diffusion

## ETHICS STATEMENT

We have read and will adhere to the ICLR Code of Ethics. This work uses only public data, involves no human subjects or personally identifiable information, and therefore does not require IRB review. Results are reported for research purposes only; we release anonymized code/configurations to support verification, and will disclose any funding sources and potential conflicts of interest upon acceptance.

## REPRODUCIBILITY STATEMENT

To support reproducibility, we release an anonymized repository with all experiment details including training/evaluation scripts, default hyperparameters, configuration files, and software/hardware environment.

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

## LLM USAGE DISCLOSURE

We used large language models (OpenAI GPT-4o and GTP-5) as auxiliary tools for grammar checking and language polishing of the manuscript. These models were not involved in research ideation, experimental design, implementation, or analysis. The authors take full responsibility for all content.

## A    RELATED WORK

### A.1    STATE-OF-THE-ART ATTRIBUTION METHODS

Attribution methods have emerged as one of the mainstream approaches for interpreting Deep Neural Networks (DNNs), owing to their ability to provide fine-grained, pixel-level explanations. We begin by introducing several commonly used attribution methods developed in recent years. The Integrated Gradients (IG) method (Sundararajan et al., 2017) addresses the issue of vanishing gradients in Saliency Map (SM) (Simonyan et al., 2013) algorithm by proposing two axioms that attribution should satisfy. IG computes attribution scores for each input feature by integrating the gradients along a straight-line path from a chosen baseline to the input. To mitigate the noise interference along the integration path, the Guided Integrated Gradients (GIG) method (Kapishnikov et al., 2021) introduces constraints on the network input and backpropagates neuron gradients to suppress irrelevant pixel attributions, retaining only those features pertinent to the predicted class. While GIG effectively reduces noise, it is primarily tailored to image-based tasks, is highly sensitive to the quality of input features, and suffers from significant computational complexity. To further improve the rationality of anchor point selection, the Boundary-based Integrated Gradients (BIG) method (Wang et al., 2021) introduces a boundary-search mechanism to achieve more accurate attribution results. BIG attempts to use adversarial examples as anchors but still relies on linear integration paths. Additionally, BIG requires gradient computations for each feature point individually, which substantially increases computational costs. In contrast, the Adversarial Gradient Integration (AGI) method (Pan et al., 2021) seeks the steepest non-linear ascent path starting from adversarial examples, thus avoiding the need for a predefined baseline as required in IG. However, AGI's effectiveness heavily depends on the quality of adversarial sample generation, and its robustness remains under debate. In addition, other adversarial attribution methods (Zhu et al., 2024b;a;c) that employ adversarial examples as attribution baselines have also been widely adopted.

### A.2    COMMONLY USED ATTRIBUTION EVALUATION METRICS

Current attribution evaluation metrics, such as the Insertion & Deletion Scores (Petsiuk et al., 2018), offer an intuitive curve-based evaluation approach. By progressively adding or removing regions with high attribution scores and observing the corresponding changes in prediction probability, these metrics aim to reflect the faithfulness of the attribution. The Deletion Score progressively occludes the pixels with the highest attribution scores (e.g., by setting them to zero or to the mean value), and plots a curve showing how the model's prediction probability decreases as the proportion of occluded pixels increases. A smaller Area under the Curve (AUC) indicates a more faithful attribution. In contrast, the Insertion Score begins with a blank or blurred baseline and progressively inserts the most important pixels based on attribution scores, observing the rate at which the model's prediction probability increases. As the Insertion and Deletion Scores often rely on filling removed regions with constant values (e.g., black or mean pixels), they are highly sensitive to distributional shifts. The resulting perturbed inputs—characterized by large artificial occlusions—can diverge substantially from the original data distribution, potentially introducing instability or bias into the evaluation outcomes. Recent research (Nieradzik et al., 2024) have sought to improve the smoothness of attribution maps by introducing adversarial perturbations or enforcing smoothness regularization during the evaluation phase. Nonetheless, such methods fall short in mitigating the distributional shift induced by masking operations, as the modified inputs often remain at the periphery of the training data distribution.

The Infidelity metric (Yeh et al., 2019) attempts to quantify attribution consistency by computing the expected mean squared error between attribution scores and the corresponding changes in model predictions under input perturbations, theoretically providing a more robust estimate. The metric selects a meaningful perturbation distribution (e.g., adding random noise to pixels or occluding a patch), and computes the expected mean squared error between the change in the model's output

and the inner product of the perturbation with the attribution. A lower Infidelity score indicates higher attribution faithfulness. It is worth noting that the experiments compute the Infidelity metric using perturbations such as 'noise baselines' and 'patch removal'. However, commonly used global patch occlusion methods often result in unrealistic images, which may compromise the reliability of the metric. Although local random noise perturbations tend to cause smaller deviations from the original data distribution, this assumption still does not fully guarantee alignment with the training distribution.

Another metric for evaluating attribution faithfulness is Sensitivity-$n$ (Ancona et al., 2017), which emphasizes the consistency between attribution scores and the model's output response. The core idea is that if certain pixels (or features) are identified as important in the attribution map, then randomly occluding these regions should lead to significant changes in the model's prediction. Specifically, Sensitivity-$n$ evaluates whether the change in the model's output is consistent with the attribution importance by randomly selecting and occluding the top-$n$ features with the highest attribution scores and measuring the resulting output variation. Unlike the Insertion & Deletion Scores, Sensitivity-$n$ does not rely on explicitly constructing a perturbation sequence or response curve, making it more computationally efficient. Moreover, it mitigates the distributional shift issue caused by unnatural occluded images. However, this metric remains sensitive to the choice of occlusion strategy—for instance, the selection of occlusion values (e.g., zero or mean replacement) can significantly influence the results. Moreover, Sensitivity-$n$ does not directly assess the causal explanatory relationship between the attribution and the model's prediction, but rather reflects local perturbation consistency. As a result, it is limited in its ability to serve as a comprehensive evaluation metric.

To mitigate the aforementioned issues, some studies have introduced blurring operations as alternatives to direct occlusion. For instance, the Meaningful Perturbation (Fong & Vedaldi, 2017) and Extremal Perturbation (Fong et al., 2019) methods optimize masks to maximize the model's output within the preserved regions. These methods produce visually more natural perturbations and reduce abrupt distributional shifts; however, the inherent blurring still preserves low-frequency features of the original image, which may lead to semantic distortions in the attribution maps. Moreover, the generated inputs are rarely encountered in the training set, thus still posing a certain risk of out-of-distribution (OOD) inputs.

Beyond blurring-based perturbations, several works have explored the use of generative models to reconstruct or in-paint the regions removed during attribution. Chang et al. (2019) generate counterfactual images by replacing selected features with samples from a conditional generative model, thereby exposing how classifiers depend on specific structures. Similarly, Agarwal & Nguyen (2020) propose filling removed pixels using a generative model to maintain perceptual realism while probing feature importance. While these approaches significantly reduce the visual artifacts of masking, the generative prior may inadvertently introduce class-supporting evidence rather than faithfully removing it, complicating deletion-based evaluation. Our FUD framework inherits the insight that perturbations should remain on the data manifold, but applies it at the level of the *evaluation metric*: any attribution method can be assessed under a unified score-based generative prior without modifying the explainer itself. Therefore, the design of attribution metrics must balance 'distributional consistency' and 'interpretability,' avoiding conclusions about attribution quality based solely on any single metric.

## B    DESIRABLE PROPERTIES OF EVALUATION PERTURBATIONS

To clarify what constitutes a reasonable modified image for attribution evaluation, we list the ideal properties that any perturbation $x'$ should satisfy:

1. **In-distribution realism.** Perturbed samples should remain on (or close to) the data manifold rather than becoming OOD artifacts. FUD enforces this via the prior term $\nabla_x \log P(x)$ and validates it through OOD-detection metrics.

2. **Preservation of retained features.** Pixels marked as "kept" must match the original input exactly. This is implemented through the hard constraint $P(\tilde{x} \mid x, M)$ and stepwise overwriting.

3. **No hallucinated class evidence.** Perturbations should not introduce features that artificially increase confidence in class $y$. FUD controls this via the corrective term $-\nabla_x \log P(y \mid x)$.

4. **Perceptual coherence.** Intermediate samples should remain structurally consistent (edges, colors, textures). This is quantified using standard perceptual metrics such as PSNR, SSIM, FSIM, and GMSD.

These desiderata generalize the assumptions underlying baseline-replacement methods. FUD operationalizes them explicitly through the posterior

$$P(x_t \mid z, \tilde{x}, M) \propto P(x_t) \, P(z \mid x_t) \, P(\tilde{x} \mid x_t, M), \tag{9}$$

whose gradient decomposition in Eq. 6 directly corresponds to properties (1)–(3), while (4) is validated empirically.

## C    PROOF OF THE $\delta-$LIKELIHOOD FOR OBSERVED PIXELS

For each pixel index $i$ we denote by $M_i = 1$ that the pixel is *observed* (i.e. must be preserved exactly in every evaluation sample) and by $M_i = 0$ that the pixel is *free* (no constraint). Hence the conditional likelihood factorises as

$$P(\tilde{x} \mid x^t, M) \;=\; \prod_{M_i=1} \delta(x_i^t - \tilde{x}_i), \tag{10}$$

where the Dirac distribution $\delta(\cdot)$ assigns non-zero density *only* when $x_i^t = \tilde{x}_i$, while the factor 1 leaves unobserved pixels unconstrained. The support of $x^t$ is therefore restricted to the *linear sub-manifold* $\left\{ x^t \in \mathbb{R}^d \; : \; x_i^t = \tilde{x}_i \text{ for all } M_i = 1 \right\}$.

**Dirac as the zero-variance limit of a Gaussian**    To justify the $\delta$-likelihood formally, consider the single–pixel Gaussian proxy

$$\mathcal{N}\big(x_i^t; \tilde{x}_i, \sigma^2\big) = \frac{1}{\sqrt{2\pi}\sigma} \, \exp\!\Big[-\frac{(x_i^t - \tilde{x}_i)^2}{2\sigma^2}\Big]. \tag{11}$$

Taking the limit $\sigma^2 \to 0$ yields $\mathcal{N}(x_i^t; \tilde{x}_i, \sigma^2) \xrightarrow[\sigma \to 0]{} \delta(x_i^t - \tilde{x}_i)$. Multiplying the Gaussian factors over all $i$ with $M_i = 1$ and letting $\sigma^2 \to 0$ produces precisely the product of Dirac distributions used above.

**Intuition**    This formulation can be understood as an extreme case of a "zero-variance Gaussian," where non-zero probability mass exists *only* when $x_{(i)}^t = \tilde{x}_{(i)}$ for observed pixels. That is, the distribution has support strictly limited to the set of values that exactly match the ground truth on known entries. For unobserved pixels where $M_i = 0$, the likelihood imposes no constraints—effectively acting as a multiplicative factor of one. As a result, the support of $x^t$ becomes a linear sub-manifold in which all observed pixels must precisely align with their true values. This induces a hard constraint in the generative process: *any valid sample must match the observed data exactly*.

**Log-likelihood and gradient**    Because $\delta(\cdot)$ is not a conventional density, its logarithm is undefined; nevertheless the *gradient* of the log-likelihood can be obtained safely via the Gaussian limit. For the vector $x^t \in \mathbb{R}^d$ we have

$$\log P(\tilde{x} \mid x^t, M) = \sum_{M_i=1} \log \delta(x_i^t - \tilde{x}_i), \tag{12}$$

$$\nabla_{x^t} \log P = \frac{M \odot (\tilde{x} - x^t)}{\sigma^2} \xrightarrow[\sigma \to 0]{} \text{a vector pointing towards infinity} \tag{13}$$

where $\odot$ denotes the element-wise product and $M \in \{0,1\}^d$ is the mask vector. As $\sigma \to 0$, the magnitude of the gradient diverges while its direction always points from $x^t$ back to the true pixel values $\tilde{x}$: the optimisation is therefore forced *instantaneously* onto the constraint manifold.

**Practical implementation**    In practice we do not apply the infinite gradient. Instead, after each diffusion update we simply overwrite the observed pixels:

$$x^t \;\leftarrow\; M \odot \tilde{x} \;+\; (1 - M) \odot x^t, \tag{14}$$

which is exactly equivalent to following the $\delta$-likelihood's gradient in the $\sigma^2 \to 0$ limit but avoids numerical instabilities.

## D    ADDITIONAL ANALYSIS OF THE SCORE FUNCTION

**Why delay the classifier gradient?**    As argued in Section 3.3.2, adding the classifier term $-\nabla_{x^t} \log P(y \mid x^t)$ too early can push off-manifold samples further away; guidance is beneficial only after the diffusion trajectory has moved close to the data manifold under the prior score $s_\theta(x^t)$.

This is because at early steps, the sample $x^t$ is still far from the classifier's data distribution, making the gradient signal from the classifier unreliable or even misleading. Delaying the classifier guidance ensures that meaningful and stable gradients are provided only when the sample is sufficiently close to the data manifold.

**Experimental protocol**    We fix the total diffusion steps at $T = 1000$ and **always** turn on classifier guidance for the last $5\%$ of those steps ($t < 50$). To study the effect across different deletion levels, we vary the mask ratio $\rho \in \{10\%, 20\%, \dots, 90\%\}$, where $\rho$ denotes the percentage of *unimportant* features removed by the attribution method before FUD starts sampling. For each $\rho$ we generate 500 evaluation samples and measure the AUROC between the retained-feature ratio and the classifier confidence.

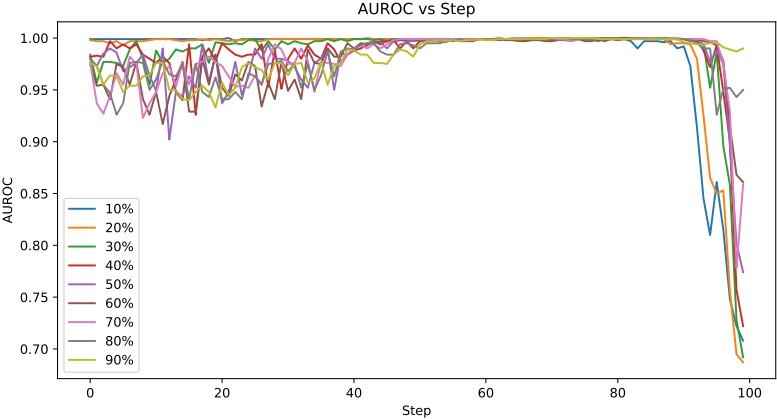

Figure 3: AUROC versus diffusion step for different **mask ratios** $\rho$ (10%–90%). Classifier guidance is enabled for the last 5% of steps (grey vertical line). All curves stabilise quickly after guidance kicks in, and the final AUROC is highest under this 5% schedule.

**Results**    Figure 3 plots AUROC versus diffusion step for six representative mask ratios. A consistent pattern emerges: AUROC remains flat while only the prior $s_\theta(x^t)$ is applied, then climbs sharply once classifier guidance begins at the $t/T = 0.05$ mark, finally saturating within 30–40 steps. Although higher mask ratios (e.g. 70–90 converge to stable curves once guidance is applied, confirming that the **"5% switch"** is robust across deletion levels.

## E    PSEUDOCODE FOR FUD EVALUATION

**Note:** In practice, we use DDIM to accelerate the sampling process. This procedure in Algorithm 1 can be viewed as a discretized version of DDPM sampling.

---

**Algorithm 1: FUD Evaluation**

---

**Input:** original image $\tilde{x}$;
*mask* $M$;
out–of–distribution *threshold* $O$;
Score Function $S_\theta(x, \varepsilon, t)$;
classifier $P(y \mid x)$;
total steps $T$, DDIM noise schedule $\{\bar{\alpha}_t\}_{t=0}^{T}$
**Output:** in-distribution evaluation sample $x^0$

$\varepsilon \sim \mathcal{N}(0, I) \; x^T \leftarrow \tilde{x} \odot M + \varepsilon \odot (1 - M)$
**for** $t = T, \ldots, 1$ **do**
    **if** $t < O$ **then**
        $\hat{\varepsilon} \leftarrow S_\theta(x^t, t)$
    **else**
        $\hat{\varepsilon} \leftarrow S_\theta(x^t, t) + \sqrt{1 - \bar{\alpha}_t} \, \nabla_{x^t} \log P(y \mid x^t)$
    $x^{t-1} \leftarrow \sqrt{\bar{\alpha}_{t-1}} \left( \dfrac{x^t - \sqrt{1 - \bar{\alpha}_t} \, \hat{\varepsilon}}{\sqrt{\bar{\alpha}_t}} \right) + \sqrt{1 - \bar{\alpha}_{t-1}} \, \hat{\varepsilon}$
    $x^{t-1} \leftarrow \tilde{x} \odot M + x^{t-1} \odot (1 - M)$
**return** $x^0$

---

## F  METRIC DEFINITIONS

Each metric in Table 6 is briefly defined below:

- **PSNR** – Peak Signal-to-Noise Ratio; measures average pixel fidelity.
- **SSIM** – Structural Similarity Index; compares luminance, contrast, and structure.
- **MS-SSIM** – Multi-Scale SSIM; aggregates SSIM over multiple resolutions.
- **FSIM** – Feature Similarity; integrates phase congruency and gradient magnitude.
- **GMSD** – Gradient Magnitude Similarity Deviation; lower values indicate fewer edge distortions.
- **HaarPSI** – Haar Wavelet–based Perceptual Similarity; focuses on multiscale edge recall.
- **VSI** – Visual Saliency–based Index; emphasises fidelity in salient regions.

Table 6: Image–quality metrics used to assess transitional samples. For GMSD we invert the score $(1 - \text{GMSD})$ when plotting the radar chart to align "higher–is–better" semantics.

| Metric | High Value Means | Low Value Means | Trend |
|---|---|---|---|
| PSNR | Low pixel distortion | Large pixel error | ↑ |
| SSIM | High structural similarity | Blurring / structure loss | ↑ |
| MS-SSIM | Multi-scale consistency | Local distortion | ↑ |
| FSIM | Sharp edges / textures | Edge and detail loss | ↑ |
| GMSD | Small gradient deviation | Edge blur, contour loss | ↓ |
| HaarPSI | Good multi-scale detail | Global blur | ↑ |
| VSI | Clear salient regions | Salient region blur / loss | ↑ |

**Comparing FUD to existing deletion baselines.** The upper sub-row of each example in Fig. 4 (*Original → Heat-map → INFD → INS/DEL → Sen-N*) visualises three widely–used deletion metrics. Despite their popularity, all three baselines exhibit conspicuous off-manifold artefacts even before **half** of the pixels are masked:

- **INFD** (third column) applies a saliency–guided Gaussian blur. At moderate deletion ratios the foreground object dissolves into low-frequency smear, but background textures remain untouched—contradicting the intended focus on "unimportant" regions.

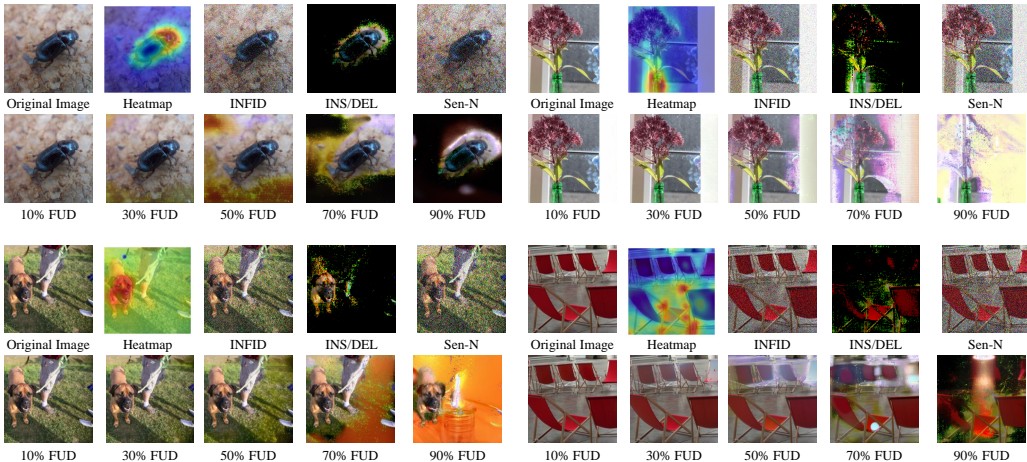

Figure 4: **Qualitative evolution of FUD.** From left to right: original image, LA (Zhu et al., 2024d) heat-map, and FUD samples after deleting $(100 - \rho)\%$ *unimportant* pixels and reconstructing the removed regions via score-based diffusion under the learned data distribution, rather than using any constant black/white filling. The apparent contrast changes therefore come from generative in-painting on the image manifold, not from hard occlusions as in INS/DEL or Sen-N. Rows correspond to four randomly–chosen validation images.

- **INS/DEL** (fourth column) literally *zeros* the unimportant features, producing unnatural black cavities that trigger premature confidence drops and confound any perceptual judgement.

- **Sen-N** (fifth column) injects pixel-wise Gaussian noise; as deletion grows the image devolves into high-frequency snow, masking true object boundaries and violating the data-manifold assumption behind the classifier.

**FUD yields natural transitional samples.** The lower sub-row shows FUD results for five retained-feature ratios $\rho = 10\%, 30\%, 50\%, 70\%, 90\%$. Three qualitative patterns stand out:

1. **Perceptual realism up to 50 %.** When $\rho \geq 50\%$ the generated evaluation images are almost indistinguishable from the originals in *both* structure and colour palette. Crucially, no "floating" fragments or unnatural voids appear, confirming that FUD keeps the trajectory on the image manifold.

2. **Smooth degradation beyond 50 %.** As the mask ratio increases to 70 – 90 %, FUD removes the object in a coarse-to-fine fashion: salient boundaries blur first, then disappear into context-aware textures. The resulting images are still globally coherent—e.g. the beetle body in Example A melts into surrounding earth tones, and the red deck-chair pattern in Example D fades without breaking symmetry.

3. **No class-switch artefacts.** Even in the extreme 90 % deletion case, FUD never hallucinates features suggestive of a different ImageNet class. This visually supports the theoretical constraint in Eq. (7) that prevents samples from crossing class manifolds.

**Implications for metric fidelity.** Because FUD maintains high perceptual quality *until* more than half of the high-attribution pixels are removed, the resulting deletion curve (cf. Fig. 6, main paper) reflects *true* model reliance on the preserved features rather than spurious artefacts. Conversely, the rapid confidence collapse observed with INS/DEL or Sen-N can be attributed to their off-manifold distortions rather than to the attribution map itself. Hence FUD provides a more faithful and interpretable evaluation of saliency methods.

Table 7: FUD, Insertion, Deletion, Sensitivity-$n$, and INFID scores for 11 representative attribution methods. Higher is better for FUD, Insertion, and Sensitivity-$n$; lower is better for Deletion and INFID.

| Method | FUD | Insertion | Deletion | Sensitivity-$n$ | INFID |
|--------|-----|-----------|----------|-----------------|-------|
| SM | 0.282534 | 0.314616 | 0.068716 | 0.141470 | 0.287042 |
| IG | 0.313328 | 0.344998 | 0.052537 | 0.126894 | 0.129003 |
| FIG | 0.208447 | 0.245956 | 0.069740 | -0.130477 | 0.277800 |
| BIG | 0.354196 | 0.435646 | 0.084743 | 0.027358 | 0.019717 |
| MFABA | 0.276959 | 0.374439 | 0.118989 | 0.034741 | 0.019622 |
| AttEXPlore | 0.276253 | 0.468133 | 0.056584 | 0.010449 | 0.019581 |
| GIG | 0.309788 | 0.332303 | 0.047974 | 0.123184 | 0.113368 |
| AGI | 0.318770 | 0.443992 | 0.057337 | 0.013355 | 0.020138 |
| ISA | 0.232271 | 0.567845 | 0.101100 | 0.023021 | 0.019569 |
| EG | 0.274612 | 0.319021 | 0.076970 | -0.007913 | 0.219567 |
| LA | 0.268251 | 0.520222 | 0.054138 | 0.019070 | 0.019574 |

## G  ADDITIONAL ANALYSIS OF WHAT FUD MEASURES

In this section we provide a more detailed discussion of what FUD actually measures and how it affects the ranking of attribution methods.

By design, a higher FUD score indicates that, when we progressively remove the features ranked as "low importance" by an attribution method, the model confidence decreases more slowly and more smoothly along the deletion path. Intuitively: (i) if an attribution method can accurately identify the truly important regions, then removing the features it considers unimportant should keep the model performance relatively high for a longer portion of the deletion trajectory, and the corresponding FUD curve should exhibit a smoother and slower decay; (ii) therefore, a higher FUD score can be interpreted, *under the constraint of staying in-distribution*, as evidence that the method is more accurate at identifying genuinely important regions.

To make the effect of FUD on the ranking more transparent, we report in Table 7 the scores of 11 representative attribution methods under five evaluation metrics: FUD, Insertion, Deletion, Sensitivity-$n$, and INFID. Here, higher values are better for FUD, Insertion, and Sensitivity-$n$, whereas lower values are better for Deletion and INFID. Gradient-path based methods such as BIG, AGI, IG and GIG achieve relatively higher FUD scores, whereas some methods that perform well under conventional Insertion/Deletion metrics (for example, variants that rely on aggressive masking or boundary attacks) move down in the ranking under FUD. Combined with the Kendall's $\tau$ analysis in Sec. 4.3.3, we observe that, under traditional INS/DEL evaluation, methods like IG and GIG can exhibit non-monotonic behaviour and even abnormal confidence increases, while under FUD their deletion curves become much smoother and closer to monotonic decay. This is consistent with the overall pattern in Table 7: FUD not only reorders existing attribution methods, but also clarifies which methods are more trustworthy when evaluated in-distribution and without relying on spurious evidence, thereby changing our empirical judgement about which family of attribution methods is more faithful.

## H  EXTENDED TABLE FOR TABLE 2

Compared to the Table 2 in the main text, we have added the results of AUPR-In here.

## I  ADDITIONAL EXPERIMENTS COMPARISON WITH PRIOR OOD-AWARE METHODS

In addition to the experiments reported in the main paper, we conducted supplementary evaluations to further validate the effectiveness of FUD against recent attribution evaluation protocols such as IDSDS (Hesse et al., 2024), ROAD (Rong et al., 2022), Distill baselines (Sturmfels et al., 2020),

Table 8: OOD detection performance for intermediate evaluation samples (higher values indicate easier detection as OOD). We report AUROC (higher means better OOD discrimination), $FPR_{95}$ (false positive rate at 95% true positive rate), AUPR-In, and AUPR-Out for a standard OOD detector distinguishing intermediate samples (OOD) from normal validation images (ID). **Bold** highlights the values indicating the **most ID-like (hardest to detect)** samples in each case. *Note:* In this context, AUROC, AUPR-In, and AUPR-Out values approaching 0.5 indicate maximal uncertainty in distinguishing between ID and OOD samples, thus reflecting increased in-distribution similarity and greater detection difficulty.

| Model | Evaluation | AUROC ↓ | $FPR_{95}$ ↑ | AUPR-In ↓ | AUPR-Out ↓ |
|---|---|---|---|---|---|
| RESNET-50 | INS/DEL | 0.8974 | 0.3603 | 0.8948 | 0.8893 |
| | Sen-N | 0.8773 | 0.5450 | 0.8579 | 0.8864 |
| | INFID | 0.7801 | 0.7720 | 0.7526 | 0.7876 |
| | FUD (Ours) | **0.6863** | **0.8317** | **0.6660** | **0.6922** |
| VIT-B/16 | INS/DEL | 0.8784 | 0.4761 | 0.8529 | 0.8894 |
| | Sen-N | 0.8781 | 0.5660 | 0.8568 | 0.8734 |
| | INFID | 0.8181 | 0.7390 | 0.7904 | 0.8119 |
| | FUD (Ours) | **0.6450** | **0.9404** | **0.5827** | **0.6812** |

and Gevaert et al. (2024). The results reinforce that FUD consistently produces in-distribution, high-fidelity transitional samples and more faithful attribution evaluations.

## I.1 PERCEPTUAL AND STRUCTURAL FIDELITY

Table 9 compares image quality metrics of transitional samples generated by different evaluation schemes. FUD achieves the best performance across nearly all metrics, indicating that its generated samples remain both perceptually and structurally closer to the natural data distribution.

Table 9: Additional perceptual/structural fidelity results. Higher values are better for PSNR/SSIM/MS-SSIM/FSIM/HaarPSI/VSI, lower is better for GMSD.

| Evaluation | PSNR ↑ | SSIM ↑ | MS-SSIM ↑ | FSIM ↑ | GMSD ↓ | HaarPSI ↑ | VSI ↑ |
|---|---|---|---|---|---|---|---|
| IDSDS Hesse et al. (2024) | 10.592 | 0.487 | 0.459 | 0.699 | 0.323 | 0.263 | 0.864 |
| ROAD Rong et al. (2022) | 18.169 | 0.385 | 0.783 | 0.728 | 0.159 | 0.594 | 0.868 |
| Distill Sturmfels et al. (2020) | 16.273 | 0.385 | 0.631 | 0.683 | 0.227 | 0.465 | 0.853 |
| Gevaert Gevaert et al. (2024) | 23.829 | 0.722 | 0.859 | 0.805 | 0.161 | 0.620 | 0.936 |
| **FUD (Ours)** | **25.200** | **0.750** | **0.780** | **0.860** | **0.124** | **0.663** | **0.946** |

## I.2 OOD DETECTION ROBUSTNESS

Table 11 reports the ability of a standard OOD detector to distinguish transitional samples from true in-distribution (ID) data. Lower AUROC, AUPR-In, and AUPR-Out, and higher FPR@95TPR indicate samples are harder to detect as OOD (i.e., closer to ID). FUD consistently yields the most ID-like samples across both ResNet-50 and ViT-B/16.

## I.3 ADDITIONAL RUNTIME ANALYSIS OF DDIM SAMPLING

To complement the perturbation analysis in this appendix, we also report the wall-clock runtime of the DDIM sampler on ViT-B/16 for different numbers of diffusion steps. As shown in Table 10, the cost grows approximately linearly with the number of steps: increasing from 10 to 100 steps raises the per-image runtime from about 0.40 s to 4.04 s. Importantly, these timings correspond to the *end-to-end evaluation* of a single image–attribution pair at the given step count, i.e., they already include the full DDIM-based perturbation process along the deletion trajectory. In our experiments we use 100 steps to obtain stable in-distribution samples, but smaller step counts (e.g., 40–60) already keep the overhead within a few seconds per image, indicating that FUD remains computationally manageable for attribution evaluation.

Table 10: Runtime of the DDIM sampler with different numbers of diffusion steps on ViT-B/16. The numbers report average wall-clock time per image (batch size 1), including the time to apply all corresponding perturbation steps.

| DDIM steps | 10 | 20 | 30 | 40 | 50 | 60 | 70 | 80 | 90 | 100 |
|---|---|---|---|---|---|---|---|---|---|---|
| Time (s) | 0.395 | 0.804 | 1.210 | 1.620 | 2.010 | 2.430 | 2.820 | 3.225 | 3.615 | 4.041 |

Table 11: Additional OOD detection results of intermediate samples.

| Model | Evaluation | AUROC ↓ | FPR@95TPR ↑ | AUPR-In ↓ | AUPR-Out ↓ |
|---|---|---|---|---|---|
| ResNet-50 | IDSDS Hesse et al. (2024) | 0.8016 | 0.5653 | 0.7907 | 0.7965 |
| | ROAD Rong et al. (2022) | 0.7123 | 0.8226 | 0.6838 | 0.7210 |
| | Distill Sturmfels et al. (2020) | 0.8226 | 0.5837 | 0.8107 | 0.8178 |
| | Gevaert Gevaert et al. (2024) | 0.7690 | 0.6552 | 0.7585 | 0.7603 |
| | **FUD (Ours)** | **0.6863** | **0.8317** | **0.6660** | **0.6922** |
| ViT-B/16 | IDSDS Hesse et al. (2024) | 0.8329 | 0.5803 | 0.8059 | 0.8358 |
| | ROAD Rong et al. (2022) | 0.7584 | 0.8000 | 0.7209 | 0.7505 |
| | Distill Sturmfels et al. (2020) | 0.8174 | 0.6795 | 0.7828 | 0.8255 |
| | Gevaert Gevaert et al. (2024) | 0.6675 | 0.9024 | 0.6201 | 0.7013 |
| | **FUD (Ours)** | **0.6450** | **0.9404** | **0.5827** | **0.6812** |

## J  L2 DISTANCE BETWEEN PERTURBED AND ORIGINAL IMAGES

To quantify how strongly our diffusion-based perturbation modifies the input, we measure the pixel-wise $L_2$ distance between the perturbed images and the original image $x$.

First, for a fixed mask ratio of $10\%$, we track the distance between the intermediate denoised samples $x_t$ and the original image $x$ over the diffusion steps. As shown in Fig. 5, the mean $L_2(x_t, x)$ starts around 80 at the initial noisy state and decreases smoothly and monotonically as the diffusion proceeds, reaching values below 5 after 100 denoising steps. The shaded band indicates one standard deviation across the evaluation set. This confirms that the optimisation gradually *reduces* the perturbation while driving the samples back toward the data manifold, instead of introducing additional distortion.

Second, we measure the final distance between the reconstructed image $x_0$ and the original image $x$ for different mask ratios. Figure 6 reports the mean $L_2(x_0, x)$ as a function of the mask ratio. The distance grows approximately monotonically with the amount of masked area: it remains small for mask ratios around $10$–$30\%$, becomes moderate for $40$–$60\%$, and only becomes large when $70$–$90\%$ of the image is removed. This shows that in the mask regimes typically used for attribution evaluation, our method perturbs the image relatively mildly, and stronger deviations occur only when a substantial portion of the content is intentionally removed.

## K  ADDITIONAL ANALYSIS OF PERTURBATION MAGNITUDE

## L  NOTATION SUMMARY FOR FUD

For clarity, we summarise the main symbols used in Section 3.3 (Faithfulness Under the Distribution) in Table 12. All symbols follow the notation used in the main text.

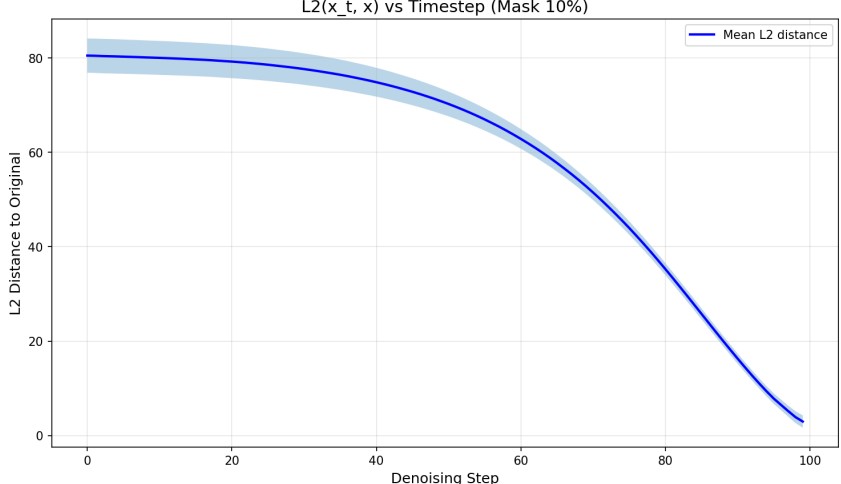

Figure 5: $L_2(x_t, x)$ as a function of the denoising step for a fixed mask ratio of $10\%$. The solid curve shows the mean distance across images; the shaded region indicates $\pm 1$ standard deviation.

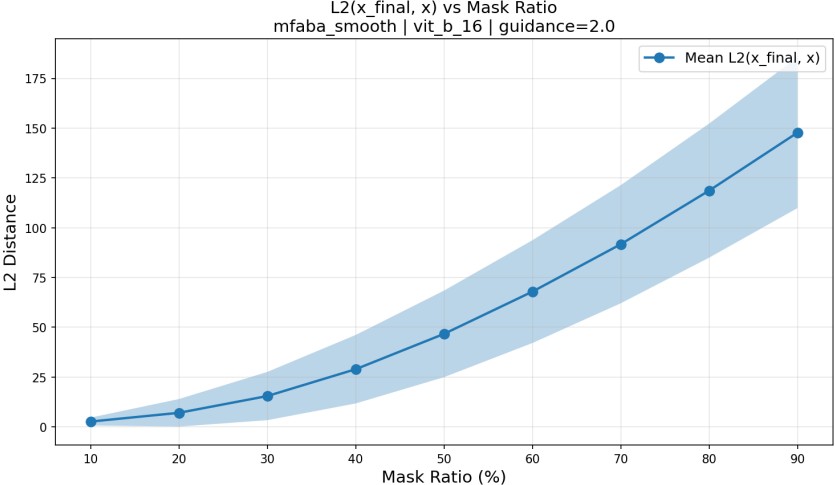

Figure 6: Final $L_2(x_0, x)$ after diffusion as a function of the mask ratio. We report the mean distance across images, with a shaded band for $\pm 1$ standard deviation.

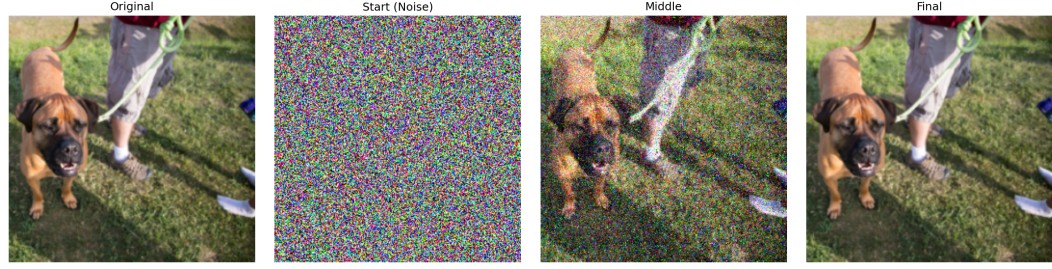

Figure 7: Qualitative "Original–Start (Noise)–Middle–Final" visualisation of the FUD denoising trajectory. From left to right: the original image $x$, the noisy masked starting point $x_T$, an intermediate denoising state $x_t$, and the final in-distribution sample $x_0$. The final image remains visually close to the original while the masked region is restored using the diffusion prior.

Table 12: Summary of notation used in the definition and derivation of FUD.

| Symbol | Description |
|---|---|
| $x$ | Original input image. |
| $\tilde{x}$ | Masked image constructed from $x$ using the deletion mask $M$ (e.g., $\tilde{x} = M \odot x + (1 - M) \odot 0$). |
| $x^t$ | Intermediate sample at diffusion step $t$ in the DDIM sampling process. |
| $x^0$ | Final in-distribution evaluation sample used for the deletion trajectory (Algorithm 1 output). |
| $y$ | Class label predicted by the classifier. |
| $f$ | Classifier under evaluation (e.g., ViT-B/16). |
| $M$ | Binary deletion mask (1 = preserve pixel, 0 = mask pixel). |
| $A(x)$ | Attribution (saliency) map for input $x$ used to construct the deletion masks. |
| $P(y \mid x^t)$ | Predictive distribution of the classifier at state $x^t$. |
| $\tilde{P}(y \mid x^t)$ | Auxiliary distribution with gradient $\nabla_{x^t} \tilde{P}(y \mid x^t) = -\nabla_{x^t} P(y \mid x^t)$. |
| $z$ | Event encoding "no additional class-$y$ evidence" (used in Eq. (1)). |
| $P(x^t \mid z, \tilde{x}, M)$ | Target conditional distribution that FUD aims to approximate. |
| $P(x^t)$ | Unconditional image prior at step $t$, approximated by the score network $s_\theta(x^t)$. |
| $P(\tilde{x} \mid x^t, M)$ | Likelihood term enforcing consistency with the observed (unmasked) pixels. |
| $P(z \mid x^t)$ | Term encoding the "no new evidence for class $y$" constraint. |
| $s_\theta(x^t)$ | Learned score network approximating $\nabla_{x^t} \log P(x^t)$ (used in Eq. 6). |
| $S_\theta(x, \varepsilon, t)$ | Score function of the DDIM sampler used in Algorithm 1. |
| $T$ | Total number of DDIM sampling steps. |
| $O$ | Out-of-distribution *threshold* on the diffusion time axis. |
| $\bar{\alpha}_t$ | Cumulative noise-schedule coefficient at diffusion step $t$. |
| $\text{FUD}(f, A)$ | FUD score of classifier $f$ under attribution map $A$. |

