~~Next, let me introduce our FUD evaluation algorithm with a brief summary for clarity.~~ The core idea of FUD is ~~to construct attribution evaluation samples that remain within the distribution while preserving the information to be evaluated. This process involves two key aspects: one is~~ **how to ensure the samples stay within the distribution**, ~~and the other is~~ **how to ensure the preservation of the evaluation information**simple: given an attribution map, we construct

perturbed samples that (i) remain on the data manifold and (ii) preserve all visible features that the attribution method identifies as important, without introducing new evidence that supports the predicted class $y$. Conceptually, this corresponds to a diffusion-style inpainting process guided by a hard mask and by a "no new evidence for class $y$" bias, so that the evaluation reflects only the contribution of the preserved features.

For completeness, we note that such perturbations are expected to satisfy several desirable properties, including in-distribution realism, exact preservation of the retained features, absence of hallucinated class evidence, and perceptual coherence. These desiderata motivate the FUD update rule, and are discussed in more detail in Appendix B.

### 3.3.2 DERIVATION

We now summarise the FUD evaluation algorithm and formalise the above intuition. The derivation addresses two questions: how to keep the perturbed samples within the data distribution, and how to preserve the information we want to evaluate.

**Within the distribution** Inspired by the SGM algorithm (Song et al., 2020), we assume that the true data distribution is $P(x)$. We denote by $x^t$ an intermediate sample at iteration $t \in \{1, 2, \ldots, T\}$, where $t$ counts the remaining update steps and each update transforms $x^t$ into $x^{t-1}$. We initialise

$$x^T = M \odot x + (1 - M) \odot \epsilon, \qquad \epsilon \sim \mathcal{N}(0, I),$$  (1)

so that the unmasked pixels follow the original image $x$, while the masked pixels are replaced by noise; in general, such an $x^t$ lies off the data manifold.

If we could obtain the gradient $\nabla_{x^t} P(x^t)$ of the true distribution at the current position, we could use gradient ascent to update $x^t$ and guide it towards regions of higher density. In practice, gradients in the input space are very sparse and the optimisation must satisfy normalisation constraints, so we instead work with the score function and update according to

$$x^{t-1} = x^t + c\,\nabla_{x^t} \log P(x^t) + \sqrt{2c}\,\epsilon, \qquad \epsilon \sim \mathcal{N}(0, I),$$  (2)

where $c$ denotes the Langevin step size and the noise term follows Song et al. (2020). We approximate the true gradient $\nabla_{x^t} \log P(x^t)$ using a learned score function $s_\theta(x^t)$, trained with the standard SGM objective

$$\theta^* = \arg\min_\theta \sum_{t=1}^{T} \lambda(t)\, \mathbb{E}_{P_{\sigma^t}(x^t)}\left[\left\|s_\theta(x^t) - \nabla_{x^t} \log P_{\sigma^t}(x^t)\right\|_2^2\right],$$  (3)

where $\lambda(t)$ and $\sigma^t$ are pre-defined hyperparameters. The score $s_\theta$ can also be learned via diffusion models such as DDPM (Ho et al., 2020). Since this paper does not focus on the training details, we refer the reader to SGM (Song et al., 2020). To

~~briefly summarize:~~ **~~once we obtain the gradient information of the distribution, we can use it to update samples via gradient steps to bring them back within the distribution~~** ~~.~~; here it is sufficient to note that, once we obtain the gradient information of the distribution, we can update samples using gradient steps to bring them back within the distribution.

**Preservation of the evaluation information**  During the evaluation of attribution methods, we progressively remove or modify ~~varying proportions of~~ features that the attribution method deems unimportant, thereby preserving ~~more~~ the features that are crucial to the model's decision-making. ~~The goal of FUD is to continue preserving the features that need to be evaluated, while simultaneously ensuring that the new evaluation samples remain~~ FUD aims to generate evaluation samples that both stay within the data distribution ~~. In this process, we must ensure two important conditions: 1) **The features that need to be evaluated must be preserved.** 2) **No features that bring the sample closer to a different class distribution are generated.** Because in the evaluation process, we aim to assess the importance of the remaining features for the model's decision on the class, if the generation process creates new features related to the class~~and preserve precisely these features. To achieve this, ~~we will be unable to distinguish whether the contribution comes from the preserved features or the newly generated ones. To satisfy this condition, we assume that we have a new distribution $\tilde{P}(y|x^t)$, the gradients of distributions $\tilde{P}(y|x^t)$ and $P(y|x^t)$ point in opposite directions, that is, $\nabla_{x^t}\tilde{P}(y|x^t) = -\nabla_{x^t}P(y|x^t)$. For ease of derivation, we~~ ~~define this event as~~ require that the features to be evaluated are kept intact, and that no newly generated features bring the sample closer to a different class distribution. Otherwise, it becomes impossible to disentangle the effect of the original preserved features from that of newly created, class-supporting evidence.

To enforce this constraint, we introduce a hypothetical distribution $\tilde{P}(y \mid x^t)$ whose gradient is opposite to that of $P(y \mid x^t)$,

$$\nabla_{x^t}\tilde{P}(y \mid x^t) = -\nabla_{x^t}P(y \mid x^t), \tag{4}$$

and we denote this "no new class evidence" bias as an event $z$. ~~At this point, the expected~~The target distribution that FUD aims to ~~learn is :~~sample from is then

$$
\begin{aligned}
P(x^t \mid z, \tilde{x}, M) &= \frac{P(z, \tilde{x} \mid x^t, M)\, P(x^t \mid M)}{P(z, \tilde{x} \mid M)} \\
&= \frac{P(z \mid x^t)\, P(\tilde{x} \mid x^t, M)\, P(x^t)}{P(z, \tilde{x} \mid M)} \;\propto\; P(x^t)\, P(z \mid x^t)\, P(\tilde{x} \mid x^t, M),
\end{aligned}
\tag{5}
$$

~~Here, $P(z, \tilde{x}|M)$ denotes a constant. If the prior distribution~~where $P(z, \tilde{x} \mid M)$ is a normalising constant and $P(x^t \mid M) = P(x^t)$ if the prior is independent of $M$~~, then it follows that $P(x^t|M) = P(x^t)$. At this point, if we compute the gradient in the same manner as before, we obtain:~~. Taking gradients yields

$$\nabla_{x^t}\log P(x^t \mid z, \tilde{x}, M) = \nabla_{x^t}\log P(x^t) - \nabla_{x^t}\log P(y \mid x^t) + \nabla_{x^t}\log P(\tilde{x} \mid x^t, M), \tag{6}$$

 with all gradients taken with respect to $x^t$.

~~The part related to $\nabla_{x^t}\log P(x^t)$ can be obtained using the $s_\theta(x^t)$ learned earlier in the paper. The $\nabla_{x^t}\log P(y|x^t)$ part can be obtained using the model corresponding to~~In this expression, the first term $\nabla_{x^t}\log P(x^t)$ is the image prior $P(x)$ (not a label vector) and is provided by the learned score network $s_\theta(x^t)$. The second term $\nabla_{x^t}\log P(y \mid x^t)$ is the classifier's input gradient with respect to $x^t$, supplied by the model whose behaviour we evaluate; in later steps, we only activate this term after $x^t$ has moved sufficiently close to the data manifold. The third term $\nabla_{x^t}\log P(\tilde{x} \mid x^t, M)$ enforces consistency on the unmasked pixels. Choosing

$$P(\tilde{x} \mid x^t, M) = \prod_{M_i=1} \delta(x_i^t - \tilde{x}_i) \tag{7}$$

ensures that these coordinates remain fixed, effectively combining the score-model prior, the ~~behavior being evaluated. It is worth noting that the computation of this term must also adhere~~

~~to the principle of remaining within the data distribution, and we will discuss the details of this later. Since $\nabla_{x^t} \log P(\tilde{x} \mid x^t, M)$ needs to fully preserve the features to be evaluated, we use $P(\tilde{x} \mid x^t, M) = \prod_{M_i=1} \delta(x_i^t - \tilde{x}_i)$ to represent it. Here, $\delta(\cdot)$ denotes the Dirac distribution. Thus, when $M_i = 1$, the gradient must be infinitely large to ensure consistency with $\tilde{x}$. Detailed properties are provided in the Appendix C. In practice~~ classifier-gradient correction, and the hard masking constraint into a single update rule in Eq. (2). In implementation, we directly replace the ~~corresponding part~~ entries of $x^t$ with $\tilde{x}$ whenever $M_i = 1$; further properties are provided in Appendix C.

~~Now, looking back at the construction of $\nabla_{x^t} \log P(y \mid x^t)$, this expression~~ The construction of $\nabla_{x^t} \log P(y \mid x^t)$ can be seen as ~~taking~~ the gradient of the input ~~sample~~ with respect to the negative cross-entropy loss~~function~~. However, ~~as we mentioned earlier, the model's behavior is only reliable for samples within the distribution. In this process, we need to ensure that $x^t$ remains within the distribution. Obviously, the initially updated~~ the model behaviour is reliable only for samples that lie within the data distribution. The initial sample $x^T$ ~~will lie outside the distribution~~is far from the manifold, and the ~~initially obtained $\nabla_{x^t} \log P(y \mid x^t)$ also lacks reference significance (The gradient information of a similar nature in SGM requires additional training of the model, sacrificing classification performance to enable it to provide effective gradient information~~ corresponding $\nabla_{x^t} \log P(y \mid x^t)$ is therefore not meaningful. In SGM-like approaches, similar gradients require additional training that sacrifices classification performance in order to obtain useful gradients on noisy samples. ~~This approach does not allow for the evaluation of extra models). Therefore, in this~~process, which is incompatible with evaluating arbitrary pretrained models. To avoid this, we initially ~~use $P(x^t|\tilde{x}, M) = \frac{P(\tilde{x}|x^t,M) \cdot P(x^t)}{P(\tilde{x}|M)} \propto P(x^t) \cdot P(\tilde{x}|x^t, M)$ to replace the distribution~~ignore the $z$-term and use

$$P(x^t \mid \tilde{x}, M) = \frac{P(\tilde{x} \mid x^t, M)\, P(x^t)}{P(\tilde{x} \mid M)} \;\propto\; P(x^t)\, P(\tilde{x} \mid x^t, M) \tag{8}$$

instead of $P(x^t \mid z, \tilde{x}, M)$, ~~so that the samples move~~ using only the score prior and mask constraint to move samples closer to the ~~distribution. In the~~ manifold. In Appendix D, we ~~use~~show empirically, using the score function to generate evaluation samples~~and find that when updating $P(x^t|\tilde{x}, M)$ for about 5%~~, that after updating $P(x^t \mid \tilde{x}, M)$ for about 5% of the remaining sampling steps, the samples begin to enter the ~~space within the distribution. At the same time, since **features close to the original class sample distribution cannot be generated**, we replace the distribution with the originally designed $P(x^t|z, \tilde{x}, M)$ at this point, and then proceed to generate~~ in-distribution region. At that point, features close to the original class distribution have not yet been newly generated, so we switch to the original target distribution $P(x^t \mid z, \tilde{x}, M)$ and continue sampling to obtain high-quality evaluation samples.

Finally, we ~~need to add some implementation details regarding the use of FUD . FUD can generate attribution~~summarise how FUD is used as an evaluation metric. FUD generates attribution–evaluation samples that ~~remain~~stay within the distribution while preserving the ~~information that needs~~features that need to be evaluated. ~~Therefore, we can follow the design of the deletion score by continuously removing unimportant features , and during this removal process, use the FUD method with consistent parameters~~We follow a deletion-style protocol: we progressively remove features deemed unimportant by the attribution method, use FUD (with fixed hyperparameters) to generate the corresponding evaluation samples ~~, then observe the changes in the model'~~at each removal level, and track the model's confidence on these samples. ~~**If the attribution algorithm can accurately assess important features, it means that evaluation samples retaining the same proportion of features will exhibit higher confidence**. The reason why we only focus on retaining~~If an attribution algorithm can accurately identify important features, ~~rather than designing two separate removal modes like insertion score and deletion score, is that~~ then evaluation samples retaining the same proportion of features will exhibit higher confidence. We only consider this "retain-important-features" direction, rather than defining separate insertion and deletion scores, because evaluating samples where only unimportant features are ~~retained is very likely to be meaningless. For example, if we want to perform the black cat and white cat classification mentioned above, where the cat's background is grass, retaining two~~kept is often not informative (for example, keeping a few background patches of grass ~~and debating which of these features~~

Table 1: Mainstream attribution evaluation metrics compared with FUD.

| Metric | Description |
| --- | --- |
| Insertion & Deletion (INS/DEL) | Measures change in model output after iteratively inserting / deleting high-score regions. |
| Infidelity (INFID) | Expected squared error between attribution-weighted perturbations and output change. |
| Sensitivity-$N$ (Sen-N) | Correlation between output change and random masking of the top-$N$ salient features. |

~~is more important for identifying the cat is unnecessary.Meanwhile, from the perspectiveof metric implementation, due to~~ in a black-cat vs. white-cat task). From an optimisation perspective, the presence of ~~$-\nabla_{x^t} \log P(y \mid x^t)$, evaluating the "unimportant" features runs counter to the optimization objective. Adversarial effects can cause instability in the evaluation~~ the $-\nabla_{x^t} \log P(y \mid x^t)$ term also means that explicitly evaluating "unimportant" features would tend to amplify adversarial effects and destabilise the metric, making it ~~impossible to distinguish whether a feature is truly unimportant or if it is due to excessive influence from $-\