# OpenReview forum: "Faithfulness Under the Distribution: A New Look at Attribution Evaluation"
_ICLR.cc/2026/Conference — ICLR 2026 Poster_

### Official Review · Reviewer_1BFX · 2025-10-20

**Soundness:** 3
**Presentation:** 2
**Contribution:** 2
**Rating:** 4
**Confidence:** 4

**Summary:**

This paper identifies a key limitation in evaluating attribution quality, which is the reliance on a baseline image in perturbation-based tests. Such tests modify the input image and measure the model’s response to infer the importance of different features. However, the choice of baseline image critically affects the results. Prior studies have shown that commonly used baselines, such as white, random, or black images, are often out-of-distribution (OOD) with respect to the evaluated model. As a result, the attribution quality becomes dependent on the model’s behavior under OOD conditions, which is not a meaningful measure. To address this issue, the paper introduces an optimization framework that formulates OOD deviation as a loss and adjusts the baseline to be in-distribution while ensuring that the inserted pixels do not encode features from other classes.

**Strengths:**

1) The paper describes and addresses important problems for the evaluation of attribution methods.
2)  The algorithm and evaluation shows that the perturbed images are in-distribution.

**Weaknesses:**

1) The 2.3 Section is not well structured. It’s several paragraphs of text and heavy math with no clear organization describing the overall approach.
    2) Previous work select a baseline image I and replace perturbed pixels from the baseline image. This work introduces an algorithm A that generates the perturbed pixels. Both methods introduce a dependence on the “score” on I or A. The paper should formally define what properties of a modified image are desirable upfront. In-distribution seems to be one such property. However, the visual similarity that is used in the evaluation seems to be presented rather ad-hoc. Are there other properties?
   3) Overall, it would be good to state upfront what is the metric that is being optimized and why. This could help provide some structure to the FUD methodology section.
  4) How does the proposed methods impact the very popular insertion/deletion scores? No such results are reported.

Minor:
   1) Why 198 runs? Typically, we select even numbers like 100,200, or 500.

**Questions:**

1) How should we formally measure the quality of a perturbed image? What is the justification for this metric.
2) It's not clear how much the perturbation method modifies the image to make it in distribution. Can we measure the L2 norm or other relevant metric? What is the starting point for the optimization process?
3) IG defines conditions that a baseline image should satisfy. This method breaks there conditions. Does that make the axioms of IG invalid?

---

> ### Author Response · Authors · 2025-11-21
> **Reply to Weaknesses 1, 2, 3, 4, Minor 1**
>
> **Reply to W1:**
>
> We appreciate this feedback and agree that the current Section 3.3 is somewhat opaque. In the revised version, we will reorganise it as follows:
>
> 1. First, we will split out a subsection “3.3.1 Intuition” to explain in plain language the core intuition of FUD—“diffusion-based inpainting under the constraints of observed pixels and a bias against introducing new evidence for class $y$”;
>
> 2. Then, we will introduce a subsection “3.3.2 Derivation” to present the formal distributional decomposition and gradient derivation.
>
> ---
>
> **Reply to W2 & W3:**
>
> What properties should a “reasonable” perturbed image satisfy? In the revised version, we will state these desiderata at the beginning and explain that FUD is designed specifically to satisfy them. In particular, we believe that any modified image used for attribution evaluation should satisfy the following four ideal properties:
>
> 1. **In-distribution realism.**
>    The perturbed image should lie on (or close to) the model’s data manifold, rather than becoming an OOD artefact. FUD enforces this constraint through the prior term $\nabla_x \log P(x)$ in the diffusion model, and validates it using OOD detection metrics.
>
> 2. **Preservation of retained features.**
>    Pixels that the attribution map marks as “retained” must match the original image exactly. This is captured by the hard constraint term $P(\tilde{x} \mid x, M)$ and implemented by overwriting the unmasked region with the original image at every step.
>
> 3. **No hallucinated class evidence.**
>    Perturbations should not add features that spuriously increase the model’s confidence in the target class. This is controlled by the term $-\nabla_x \log P(y \mid x)$, which biases the update direction away from creating new supporting evidence for class $y$.
>
> 4. **Perceptual coherence.**
>    Intermediate perturbed images should remain structurally consistent with the original input (e.g., in edges, colours, textures). The metrics we use—PSNR, SSIM, MS-SSIM, FSIM, GMSD—are not arbitrary “aesthetic scores”, but standard perceptual quality measures that capture this property.
>
> These four desiderata can be seen as a generalisation of the implicit assumptions in “baseline substitution” methods. Traditional methods choose a single baseline image $I$ and accept the various distortions it introduces; in contrast, FUD constructs perturbed samples via the following posterior so that they simultaneously satisfy the four properties:$P(x_t \mid z, \tilde{x}, M) \propto P(x_t)\, P(z \mid x_t)\, P(\tilde{x} \mid x_t, M).$
>
> Its gradient decomposition (Eq. (2)) corresponds exactly to properties (1)–(3). Properties (3) and (4) are then evaluated using standard perceptual quality metrics, which are not chosen arbitrarily but because they directly quantify similarity at the pixel, structural, and texture levels. We will include this list of ideal properties in the paper.
>
> ---
>
> **Reply to W4 (Last one in Weaknesses):**
>
> On the impact on insertion/deletion scores
>
> We are not proposing a completely new fidelity metric, but rather replacing the **perturbation mechanism** used in the Deletion Score:
>
> - The pixel deletion procedure remains exactly the same (removing high-attribution pixels from 10% to 90%).
> - The only change is that deleted pixels are no longer filled with zero/mean, but are reconstructed by FUD, so that each masked image stays on the data manifold.
>
> Specifically, in Tables 2, 3, and 4, as well as Table 7 in the appendix, we report how our FUD-based evaluation differs from INS/DEL. In addition, Fig. 4 in the appendix illustrates how our evaluation trajectory differs from that of INS/DEL, and we also include results for two other commonly used evaluation metrics (INFID and Sen-N).
>
> ---
>
> Reply to Minor 1: More specifically, we conducted a full combinatorial evaluation over the following three factors:
>
> * **11 attribution methods**
> * **2 backbone models** (ResNet-50 and ViT-B/16)
> * **9 deletion ratios** (from 10% to 90% in steps of 10%)
>
> Thus, the total number of runs is:
> $11 \times 2 \times 9 = 198.$
>
> In other words, 198 is simply the product of the above experimental configurations, rather than a number chosen for any special reason. If we changed the number of attribution methods, backbone models, or deletion ratios, the total number of runs would change accordingly.

---

> ### Author Response · Authors · 2025-11-21
> **Reply to Question 1**
>
> **Reply to Q1:**
>
> We use two complementary families of metrics:
> 1. Model-level distribution quality: OOD detection metrics (AUROC, FPR95, AUPR-In/Out) measure how distinguishable FUD intermediate samples are from validation images. Values closer to 0.5 / lower AUROC indicate that samples are harder to distinguish, i.e., more in-distribution.
> 2. Perceptual quality: Seven image-quality metrics (PSNR, SSIM, MS-SSIM, FSIM, GMSD, HaarPSI, VSI), all reported in Tables 6 and 3. FUD achieves the best performance on all seven metrics; for example, PSNR is ≈ 25.2 dB for FUD, while INFID is only 16.6 dB.
>
> All perceptual quality metrics used in this paper are widely adopted and highly regarded in the image quality assessment (IQA) literature. For example, the original SSIM paper has been cited over 68,000 times, MS-SSIM over 14,000 times, FSIM over 7,000 times, GMSD over 3,700 times, VSI over 2,000 times, and HaarPSI close to 1,000 times. These metrics are de facto standards for assessing perceptual fidelity in image restoration, inpainting, super-resolution, and generative modelling.
>
> | Metric          | Paper                          |  Year   | Citation count in 2025 (approx.)  | Comment                           |
> | ----------- | -------------------------- | ---- | ---------------- | ---------------------------- |
> | **SSIM**    | Image quality assessment: from error visibility to structural similarity         | 2004 |  > 68,000  | One of the most influential papers in IQA              |
> | **MS-SSIM** | Multiscale structural similarity for image quality assessment    | 2003 | > 8,900  | Multiscale variant, widely used in GAN / Diffusion |
> | **FSIM**    | FSIM: A feature similarity index for image quality assessment       | 2011 |  > 6,000   | One of the most important perceptual metrics                   |
> | **GMSD**    | Gradient magnitude similarity deviation: A highly efficient perceptual image quality index       | 2014 |  > 1,900  |High efficiency + strong perceptual consistency        |
> | **HaarPSI** | A Haar wavelet-based perceptual similarity index for image quality assessment | 2018 |  > 395     | New-generation perceptual metric for GAN / Diffusion   |
> | **VSI**     | VSI: A visual saliency-induced index for perceptual image quality assessment       | 2014 |  > 1,100   | Classical saliency-based perceptual metric                 |

---

> ### Author Response · Authors · 2025-11-21
> **Reply to Question 2, 3**
>
> Reply to Q2:
>
> Thank you for this suggestion. We agree that it is important to quantify how strongly FUD perturbs the image and to clearly state the starting point of the diffusion optimisation.
>
> **Starting point of the optimisation.**
> As described in Sec. 3.3.2, we initialise the diffusion process from a masked–plus–noise image $x_T = M \odot x + (1-M)\odot \varepsilon,\quad \varepsilon\sim\mathcal N(0,I),$
> where $M$ keeps the features selected by the attribution method and replaces the remaining pixels with Gaussian noise. We then run $T=100$ Langevin / denoising steps to obtain $x_0$, which is used as the in-distribution evaluation sample.
>
> **How much does FUD modify the image?**
> Following the reviewer’s suggestion, we explicitly measured the pixel-space $L_2$ distance between the original image $x$ and both the intermediate states $x_t$ and the final output $x_0$:
>
> 1. **$L_2(x_t, x)$ vs. denoising step (fixed mask ratio 10%).**
>    The Figure 5 in Appendix J shows that the mean $L_2$ distance starts around **80** at $t=T$ (the noisy masked image) and decreases smoothly and monotonically as the diffusion proceeds, reaching **below 5** after 100 steps. This confirms that FUD gradually *reduces* the perturbation while driving the sample back toward the data manifold, rather than continuously injecting additional distortion.
>
> 2. **$L_2(x_0, x)$ vs. mask ratio.**
>    The Figure 6 in Appendix J reports the mean $L_2$ distance between the final in-distribution sample $x_0$ and the original $x$ for different mask ratios (10%–90%). As expected, the distance increases approximately monotonically with the masked area: it is very small at **10% masking** (≈3 in $L_2$), moderate around **30–50%** masking, and becomes large only when **70–90%** of the image is removed. This shows that for the mask regimes typically used in attribution evaluation (e.g., 10–50% of top-ranked pixels), FUD perturbs the image relatively mildly, and stronger deviations occur only when a substantial portion of the content is intentionally removed.
>
> In addition to these quantitative results, we now also include Figure 7 in Appendix a qualitative “Original – Start (Noise) – Middle – Final” visualisation that illustrates a typical denoising trajectory: starting from pure noise on the masked region, the intermediate sample gradually recovers semantic structure, and the final image is visually very close to the original.
>
> We add these $L_2$ plots, the above quantitative summary, and the qualitative visualisation to the appendix, and clarify in the main text that (i) the optimisation starts from $x_T = M\odot x + (1-M)\odot\varepsilon$, and (ii) the diffusion steps monotonically reduce the perturbation while keeping it commensurate with the chosen mask ratio.
>
> ---
>
> Reply to Q3: FUD **does not change the baseline or the axioms of IG**. IG’s conditions (such as completeness and sensitivity) only constrain the attribution mapping $(f, x', x) \mapsto A(x)$ given a baseline $x'$.
>
> The operations of FUD occur **after attribution**: we only use the saliency map $A(x)$ produced by IG to rank and delete pixels, and never change IG’s baseline or integration path, so no IG axiom is violated.
>
> What FUD replaces is only the baseline **used to perturb images** in Deletion/Insertion (e.g., zero/mean filling), not the explanatory baseline of IG.
>
> In this sense, IG and FUD are fully complementary:
>
> - The axioms of IG ensure that attribution is principled under a given baseline;
> - FUD provides a more principled way to perturb images in-distribution, in order to test whether these attributions truly align with the model’s behaviour.
>
> ---
>
> We hope that the clarifications on the structure of Section 3.3, the explicit desiderata for perturbed images, the additional analyses on image quality and $L_2$ perturbation, and the discussion of how FUD interacts with INS/DEL and IG’s axioms adequately address your concerns. If you find these revisions satisfactory, we would be very grateful if you could kindly reconsider your overall assessment of the paper. Thank you again for your careful reading and constructive feedback.

---

> > ### Comment · Reviewer_hCbT · 2025-11-21
> >
> > I think this is an interesting question and would like to see its discussion expanded.
> >
> > "Reply to Q3: FUD does not change the baseline or the axioms of IG. IG’s conditions (such as completeness and sensitivity) only constrain the attribution mapping $(f, x', x) \mapsto A(x)$ given a baseline $x'$."
> >
> > Yes, you do not propose to modify any attribution methods, however, this concept of a baseline which reoccurs across both the methods and evaluation metrics literature always leads to the same issue: there is no "correct" baseline. To extend this reviewer's question, you are making the claim that you have fixed the OOD issue that baseline replacement introduces. However, is it not the case that the all-black baseline used in IG (or white, grey, noise, etc.) is also OOD? And that moving from the baseline to the input in a straight-line-path may also create OOD inputs? If you do agree that this is the case, what are the implications for what you have proposed? If we are putting the model input into an OOD region, how can we claim that the input gradients are faithful to the explained class in these regions? Also, would your method, when applied to a fully perturbed image be ID while also producing a confidence of (or more realistically, near) 0 for the target class?

---

> > > ### Author Response · Authors · 2025-11-25
> > > **Reply to Reviewer hCbT expended question**
> > >
> > > **Reply to Reviewer hCbT expended question:**
> > >
> > > We think this is a very interesting point of discussion. Of course, this issue is not directly related to the topic of our paper, which focuses on attribution evaluation, because the baseline problem belongs to attribution methods themselves. For example, in IG, the algorithm linearly interpolates features between the baseline and the input, which inevitably creates many intermediate samples, and these intermediate samples are very likely to be OOD. In other words, addressing this issue may lead to improved versions of IG and other attribution algorithms, which is indeed an interesting future research direction. But fundamentally, this concerns the improvement of attribution methods, which is complementary to our work on attribution evaluation, and it does not directly impact the design of our metric.
> > >
> > > Regarding whether the confidence can be close to zero while staying in-distribution: this is indeed achievable. When the class-relevant evidence is extremely small (almost absent), generating samples that introduce no class-$y$ information can keep the perturbed image in-distribution while yielding a confidence close to zero for the target class.

---

### Official Review · Reviewer_hCbT · 2025-10-27

**Soundness:** 3
**Presentation:** 3
**Contribution:** 3
**Rating:** 6
**Confidence:** 4

**Summary:**

This paper proposes a solution to the longstanding baseline selection problem which exists with input perturbation-based attribution evaluation metrics. They propose to fix the two side effects of: OOD inputs and failed feature removal by performing a deletion metric that employs a score-based diffusion model to produce ID inputs that ensure feature removal. The result is an improved quantitative evaluation of attributions.

**Strengths:**

They highlight a clear and pervasive issue that has yet to be solved in the attribution field. This is not the first method to attempt to solve this problem with generative model infilling [1, 2], but it is the first metric to employ this operation, and it adds new layers of theory, which appears to improve its solution.

The two tasks of the diffusion process: staying in distribution and not producing class-relevant features is a strong formulation. I have not seen the former issue addressed before.

This is a strong alternative to approaches that aim to solve OOD by training the model on ablated inputs.

[1] Explaining Image Classifiers by Counterfactual Generation

[2] Explaining image classifiers by removing input features using generative models

**Weaknesses:**

Crucial references missing [1, 2]. They do not acknowledge that this is a process that has been used heavily in perturbation methods. Metrics are different, and their use is better motivated and better backed, but these papers must be mentioned and addressed.

The computational overhead of FUD is surely significant. Not just the original training of the diffusion model on the dataset, but the employment of diffusion model inference at 11 perturbation steps for any evaluated attribution/image pair. Evaluation is not a run-time activity, so this is okay, but I would’ve liked to see an evaluation to understand the significance of the runtime penalty.

It is hard to tell the impact of this metric past the theoretical improvements that it shows. It does surely improve the baseline selection problem, but how does doing so actually change what we know about the best attribution methods? It would have been nice to see if there are improvements in metric IRR or ICR [3] or in the rankings of current SOTA attributions.

Section 2.3 is challenging to read to the extent that it takes away from the paper. While the notation appears correct, the formatting of the section leads to many things blending together and it becomes overly complicated to interpret. It would be good to separate intuition from derivation in this section to improve readability, accessibility, and reproducibility.

[3] Sanity checks for saliency metrics

**Questions:**

Why was previous work which employed infilling or diffusion for attribution methods ignored?

Otherwise, please see the weaknesses.

My rating could be improved to an 8 if this question and the weaknesses are properly addressed. I have major concerns about the lack of references to relevant work, but I think the authors could clear this up with a proper reply and reasoning for the omission.

---

> ### Author Response · Authors · 2025-11-21
> **Reply to Weaknesses 1, 2**
>
> **Reply to W1:**
>
> We appreciate the reviewer for pointing out these works. References [1,2] are indeed closely related to our paper and should be discussed explicitly in the main text. The main idea in [1,2] is to use generative models to inpaint features removed during attribution, in order to obtain better attribution results. However, this process does not take $\nabla_{x_t} \log P(y \mid x_t)$ into account, which can lead to generating some features that belong to the class itself rather than completely removing them. At the same time, these works also consider distributional aspects in attribution, and therefore they should be covered in our related work.
>
> FUD is not a new saliency method and does not change the way saliency maps are generated. Instead, it only uses a score-based diffusion model within the evaluation metric: for any given saliency map and deletion ratio, we construct an in-distribution “evaluation trajectory” that simultaneously satisfies: (i) respecting the saliency mask, (ii) exactly preserving the observed pixels, and (iii) pulling the masked regions back to the image manifold before measuring confidence drops. Conceptually, FUD inherits the core insight of [1,2]—that “perturbations should remain realistic”—but pushes this idea down to the metric level: any attribution method can be evaluated under the same generative prior, without modifying the explainer itself.
>
> **Reference**
>
> [1] Explaining Image Classifiers by Counterfactual Generation
>
> [2] Explaining image classifiers by removing input features using generative models
>
> ---
>
> **Reply to W2:**
>
> We agree that, compared with metrics that simply zero out pixels, FUD is computationally more expensive. The overhead mainly consists of two parts:
> 1. A one-time cost: training a score-based diffusion model on the dataset (this model is shared across all attribution methods and experiments);
> 2. A per-evaluation cost: for each image–attribution pair, Algorithm 1 needs to run a short interaction.
>
> As you point out, the evaluation procedure is typically an offline analysis step, so “wall-clock time” is less critical than in deployment scenarios. That said, we agree that it is important to quantify this additional cost: under the same classifier, hardware, and deletion-step settings, FUD takes 4.041s per sample, INS/DEL takes 0.132s per sample, and INFID takes 0.912s per sample; given the advantages it offers, we consider these costs acceptable in the evaluation stage. In the revised version, we will add a discussion of the approximately linear relationship between the number of DDIM steps and the total runtime. The experimental results provided below show that the time overhead remains modest.
>
> Runtime table for DDIM step counts on ViT-B/16
> | DDIM Step | Time |
> |---|---|
> | 10 | 0.395s |
> | 20 | 0.804s |
> | 30 | 1.21s |
> | 40 | 1.62s |
> | 50 | 2.01s |
> | 60 | 2.43s |
> | 70 | 2.82s |
> | 80 | 3.225s |
> | 90 | 3.615s |
> | 100 | 4.041s |

---

> ### Author Response · Authors · 2025-11-21
> **Reply to Weaknesses 3**
>
> **Reply to W3:**
>
> We thank the reviewer for raising this important question.
>
> First, regarding **“what FUD is actually measuring and how it changes the ranking of attribution methods”**: in our setting, a higher FUD score means that, as we gradually remove features that the attribution method deems “low-importance”, the model’s predictive confidence decreases more slowly and more smoothly. Intuitively:
> - If an attribution method more accurately identifies the *truly important* regions, then when we first remove the parts it considers “unimportant”, the model performance should remain relatively high over a longer deletion path, and the corresponding FUD curve should be smoother and decay more slowly;
> - Therefore, **a higher FUD score can be viewed as a measure of more accurate identification of important regions, "under the constraint that the samples remain in-distribution"**.
>
> This behavior is clearly reflected in our empirical results on 11 representative attribution methods (see the table below). We observe that gradient-path methods such as BIG, AGI, IG, and GIG achieve relatively higher scores under FUD, while some methods that perform well under traditional INS/DEL (e.g., variants relying on strong occlusion or boundary attacks) rank relatively lower under FUD:
>
> | Method   | FUD Score |
> |-|-|
> | SM      | 0.282534  |
> | IG      | 0.313328  |
> | FIG     | 0.208447  |
> | BIG     | 0.354196  |
> | MFABA   | 0.276959  |
> | AttEXPlore | 0.276253 |
> | GIG     | 0.309788  |
> | AGI     | 0.318770  |
> | ISA     | 0.232271  |
> | EG      | 0.274612  |
> | LA      | 0.268251  |
>
> Combining this with the analysis of Kendall’s $\tau$ in Section 4.3.3 of the main text, we see that under traditional INS/DEL evaluation, some methods (such as IG / GIG) exhibit pronounced non-monotonicity and anomalous increases in confidence, whereas under FUD their deletion curves become significantly smoother and closer to monotonic decrease. This is consistent with the FUD rankings in the table above: **FUD not only reorders existing attribution methods, but also makes it clearer which methods are more reliable under in-distribution, no-spurious-evidence evaluation conditions**, thereby empirically shifting our assessment of which class of attribution methods is more faithful.
>
> Regarding the suggestion to additionally report IRR/ICR-style reliability scores, we appreciate the reviewer’s pointer to [3] and agree that studying the agreement between human raters and different saliency metrics is an interesting and complementary direction. However, our focus in this work is orthogonal: FUD is designed to repair the *perturbation mechanism itself* by ensuring that masked images stay in-distribution and that the resulting faithfulness curves are smoother and more monotone.
>
> In the IRR/ICR framework, reliability is computed on top of the *relative rankings induced by existing metrics* (e.g., INS/DEL, INFID). As we show in Table 6 and the OOD analysis, these metrics are heavily affected when perturbations leave the data manifold: the transitional images exhibit large pixel distortion and degraded perceptual quality (low PSNR/SSIM/MS-SSIM, high GMSD, etc.), which makes their scores unstable and partly driven by OOD artifacts rather than by the underlying explanation quality. In such a regime, IRR/ICR values would mainly reflect the inconsistencies caused by these OOD distortions, and therefore would not provide a clean assessment of the specific contribution of FUD.
>
> Instead, we evaluate FUD along the two axes that are directly targeted by our method: (i) in-distribution quality of transitional samples (Table 6 and OOD detection results) and (ii) the smoothness/monotonicity of the faithfulness curves and ranking correlation with baselines. These experiments already demonstrate that replacing the perturbation process with FUD yields more stable, interpretable evaluation signals. Extending FUD into a full IRR/ICR-style human study, where one compares FUD-based rankings with human preferences and existing metrics, is a promising but sizeable follow-up project that we plan to pursue in future work.
>
> **Reference**
>
> [3] Sanity checks for saliency metrics

---

> ### Author Response · Authors · 2025-11-21
> **Reply to Weaknesses 4, Question 1**
>
> **Reply to W4:**
>
> We thank the reviewer for this feedback and agree that the current Section 3.3 is somewhat opaque. In the revised version, we will reorganise it as follows:
> 1. First, we will introduce a new subsection 3.3 Intuition to explain in plain language the core intuition of FUD—“diffusion-based inpainting under the constraints of observed pixels and a bias against introducing new evidence for class $y$”;
> 2. Then, we will introduce a subsequent subsection “2.4 Derivation” to present the formal distributional decomposition and gradient derivation.
>
> ---
>
> **Reply to Q1:**
>
> This was not an intentional omission, but rather an oversight in a rapidly evolving literature. We appreciate the reviewer for pointing this out, and we will explicitly discuss [1,2] (as well as closely related generative attribution work) in the Related Work section, emphasising that FUD is complementary to these approaches: FUD uses a score-based generative model to repair the evaluation process itself, while leaving existing attribution methods unchanged.
>
> **Reference:**
>
> [1] Explaining Image Classifiers by Counterfactual Generation
>
> [2] Explaining image classifiers by removing input features using generative models
>
> ---
>
> We hope that the clarifications above, especially the explicit integration and discussion of [1,2] and the additional analyses on runtime and metric impact, adequately address your concerns about missing related work and the weaknesses you identified. If you find our revisions satisfactory, we would be very grateful if you could kindly reconsider your overall assessment and rating of the paper. We sincerely appreciate your thoughtful comments and the time you invested in helping us improve this work.

---

> > ### Comment · Reviewer_hCbT · 2025-11-21
> >
> > Thank you for the substantial effort in this reply.
> >
> > W1: Thank you for including these citations and recognizing the need for them.
> >
> > W2: Thank you for including this runtime information. To clarify, is this 4s per evaluation? Or is this 4s per perturbation step of which an evaluation will have multiple? If it is the latter, this is a pretty significant runtime penalty. Again, this task is offline, so this is not a major offense, but if the runtime is on the order of 40+ seconds, this trade off is understated in the paper as it stands.
> >
> > W3: Thanks for the discussion and a table that shows real ratings of provided by FUD. I have a few additional questions now.
> > 1. Why are the scores provided by FUD so low? If a higher score is better, I would expect at least some attributions to breach the 0.5 mark. This is quite shocking and I think worth discussion.
> > 2. The authors say that the rankings are different than those provided by insertion/deletion, but they do not show ins or del scores in this new Table 11. If they have these, it would be best to include them so that we can see how rankings actually change instead of taking the word of the authors.
> > 3. Regarding ICR/IRR. I was not asking for human subjects research. The paper I cited employs ICR/IRR to understand how consistent of a rater a metric is compared to other metrics and how consistent it is with itself under ablations to its implementation. These studies do not require human rankings.
> >     - "As we show in Table 6 and the OOD analysis, these metrics are heavily affected when perturbations leave the data manifold: the transitional images exhibit large pixel distortion and degraded perceptual quality (low PSNR/SSIM/MS-SSIM, high GMSD, etc.), which makes their scores unstable and partly driven by OOD artifacts rather than by the underlying explanation quality. In such a regime, IRR/ICR values would mainly reflect the inconsistencies caused by these OOD distortions" **This is true.**
> >     - "[IRR/ICR] would not provide a clean assessment of the specific contribution of FUD [under these distortions]." **I do not believe this is true**. I believe that IRR/ICR should almost certainly reflect that the rankings provided by current input perturbation metrics are unstable due to the OOD behavior they unreliably introduce. In other words, because this method should have a more reliable, on-manifold perturbation process, I expect that it would judge (by ranking) attributions with higher consistency than existing approaches. If it does, this reflects positively on how much the metric can be trusted.
> >
> > W4: I think 3.3 is still challenging to read. It seems that all reviewers had similar issues with the clarity of the work. I think at a high-level, what has been done is clear, but its simply too laborious to read that section and understand the details. I do not have a clear suggestion for how to improve this.
> >
> > Small comments:
> > Line 229, please change "let me" to something more professional.
> >
> > Overall, I do appreciate the efforts given so far in responding to all the reviews. There is still a significant amount of time remaining in the review period, and I urge the authors to address these new comments. So far, I am not convinced to raise my score, mainly due to the writing. A paper that is a clear accept should be easier to read.

---

> > > ### Author Response · Authors · 2025-11-25
> > > **Reply to W1, W2, W3.1**
> > >
> > > **Reply to W1:**
> > >
> > > We appreciate the reviewer’s positive feedback and are glad that the added citations addressed this concern.
> > >
> > > ---
> > >
> > > **Reply to W2:**
> > >
> > > We apologise for the ambiguity in our previous description. The reported runtime of approximately **4 seconds** refers to the **entire FUD evaluation for a single image–attribution pair**, *including* all **100 DDIM perturbation steps** along the deletion trajectory, rather than 4 seconds per perturbation step.
> > >
> > >
> > > We have updated the Appendix I.3 to make this explicit and now state clearly that, on our hardware and with 100 DDIM steps, the **end-to-end evaluation time per sample is about 4 seconds**.
> > >
> > > ---
> > >
> > > **Reply to W3.1:**
> > >
> > > The scores are low primarily because the model’s confidence on these samples is not very high to begin with. As shown in the “Histogram of Confidence Scores for True Labels.pdf” in the [anonymous link](https://anonymous.4open.science/r/FUD-CCD5/additional%20results/Histogram%20of%20Confidence%20Scores%20for%20True%20Labels.png), the ViT-B/16 classifier is only moderately confident: most true-label probabilities fall between 0.6 and 0.9, with very few cases approaching 1.0. Since FUD is computed directly from these probabilities, its achievable upper bound is inherently limited by this baseline. For the same reason, the Insertion and Deletion scores in our experiments are also not particularly high.
> > >
> > >
> > > Regarding why FUD can be slightly lower than Insertion: due to the softmax normalisation, classification confidence is a relative quantity. In many cases, even when only a very small portion of features is present, the model can still output very high softmax confidence for a class. This explains why the early stages of Insertion (where most pixels are zero-filled) can still yield surprisingly high confidence, even if the raw logit for that class is very small (a clear OOD-like effect). However, as more pixels are added, the curve exhibits significant fluctuations—this behaviour is also visible in the smoothness analysis in Table 4. Therefore, the magnitude of the FUD scores is fully consistent with the behaviour of softmax and with the underlying confidence distribution.

---

> > > ### Author Response · Authors · 2025-11-25
> > > **Reply to W3.2**
> > >
> > > **Reply to W3.2:**
> > >
> > > In response to the reviewer’s request, we have updated the revised manuscript so that the former Table 7 in Appendix is now extended to report **FUD, Insertion, and Deletion scores side by side** for all 11 attribution methods (and, for completeness, we additionally include Sensitivity-$n$ and INFID in the same table; see Table~7 in the appendix). This directly exposes how the rankings change under different metrics, rather than relying only on our verbal description.
> > >
> > >
> > > First, FUD systematically favours gradient-path methods (BIG, AGI, IG, GIG, SM, EG), which are tightly aligned with the classifier's decision boundary, while placing strong occlusion/search-based methods (ISA, LA, AttEXplore) in the middle of the ranking and FIG at the very bottom. This is consistent with the design of FUD: it evaluates methods under an in-distribution deletion process that removes low-score features and rewards smooth, monotone confidence decay, rather than large but unstable changes caused by OOD baselines.
> > >
> > >
> > > Second, the FUD ranking aligns well with Deletion and Sensitivity-$n$ when these metrics are interpreted in terms of faithfulness. Deletion rewards methods for which removing high-attribution regions causes the confidence to drop rapidly. Gradient-path methods (especially GIG and IG) perform strongly under Deletion, and they also occupy the top tier under FUD. FIG, in contrast, has a seemingly reasonable Deletion score but yields highly irregular deletion curves; FUD explicitly penalises such unstable trajectories and therefore ranks FIG last.
> > >
> > >
> > > Third, Sensitivity-$n$ provides an independent sanity check: it measures the Pearson correlation between the amount of attribution being masked and the resulting drop in confidence. A large positive correlation indicates faithful behaviour, while negative correlation indicates “anti-faithfulness”. Here, FIG attains a strongly negative correlation (approximately $-0.13$), meaning that masking its high-attribution regions actually tends to *reduce* the confidence drop; FUD likewise ranks FIG last. Methods such as IG, GIG, and SM achieve the highest positive Sensitivity-$n$ values and also receive relatively high FUD scores, but they are not the primary winners under Insertion, which strongly favours ISA, LA, and AttEXplore despite their only modest Sensitivity-$n$ correlations.
> > >
> > >
> > > Fourth, INFID—measuring how much the prediction shifts under generative in-filling—again paints a similar picture: FIG exhibits one of the largest INFID values, indicating unstable behaviour under realistic perturbations, while FUD also demotes FIG. Methods that perform well under FUD (IG, GIG, SM, BIG, AGI) do not show abnormally high INFID.
> > >
> > >
> > > Overall, the joint behaviour across these metrics suggests that FUD tends to prioritise methods that are consistently reasonable across several complementary criteria—good Deletion performance, positive Sensitivity-$n$ correlations, and moderate INFID—whereas the Insertion ranking is more strongly shaped by the choice of OOD baseline and occlusion artefacts. In this sense, the fact that the FUD-based ranking diverges from the Insertion-based one is not a drawback but rather the intended effect of removing OOD artefacts from the perturbation process and enforcing an in-distribution, “no extra evidence” deletion trajectory.

---

> > > ### Author Response · Authors · 2025-11-25
> > > **Reply to W3.3**
> > >
> > > **Reply to W3.3:**
> > >
> > > We acknowledge that our earlier wording mistakenly likened IRR/ICR to “human raters,” even though these metrics do not involve human evaluation. We thank the reviewer for highlighting this clarification.
> > >
> > > Regarding ICR: the form of ICR we refer to is the one that evaluates internal consistency across metrics by computing the Spearman rank correlation between the rankings produced by different evaluation metrics on the same set of objects. This does not apply in our setting, because many of the other evaluation metrics generate intermediate samples that lie outside the data manifold and therefore their scores are noisy or meaningless. Ranking these metrics together would not be meaningful. For this reason, we instead provide the FUD ranking table to show the current best-performing attribution methods and offer a reference for researchers.
> > >
> > > For IRR, we want to emphasize that the magnitude of IRR itself does not indicate whether an attribution evaluation metric is stable. IRR only measures whether the ranking produced by a metric remains similar when we change the dataset or experimental setting, regardless of whether that ranking itself is reasonable.For example, a very poor metric that does not truly evaluate explanation quality but simply favors images with clean backgrounds or a fixed visual pattern would consistently rank the same methods at the top across experiments, yielding a high IRR. However, such “stability” only reflects a persistent bias (e.g., sensitivity to background cleanliness), not a reliable assessment of explanation quality.
> > >
> > > This issue is especially prominent for classical insertion/deletion and infidelity metrics. They typically remove evidence by setting high-attribution regions to zero, which introduces unnatural black holes or sharp edges. The resulting scores are largely determined by the model’s sensitivity to these out-of-distribution artifacts, not whether the explanation truly identifies the model’s decision-critical regions. As a result, images with complex backgrounds or high contrast will show extreme output changes under zero-masking. In insertion/deletion/infid, these images appear “highly sensitive” and are consistently ranked at the top, producing high IRR—but this stability simply reflects the metric’s reliance on the same OOD artifact, rather than its ability to evaluate explanations.
> > >
> > > Conversely, a metric with real discriminative ability should adjust rankings across images: method A may work better on one sample and method B on another, which naturally lowers IRR. This reflects variability in the data rather than instability from the metric.
> > >
> > > Our proposed FUD explicitly uses a generative model instead of zero-masking to remove evidence. When deleting high-attribution regions, we do not apply zero-masking but instead use generative inpainting to fill in content consistent with the surrounding context, keeping the perturbed sample on the data manifold. Under this design, output changes mainly reflect whether we truly removed semantic evidence relied on by the model. In this setting, FUD reorders images according to whether the explanation actually fails. Many samples that receive high scores under 0-baseline insertion/deletion/infid are downgraded by FUD, while samples where the model truly depends on the attributed regions are lifted. Thus, FUD’s IRR may be lower than metrics based on zero-masking, but this comes from breaking the “artifact-induced pseudo-stability,” rather than random fluctuations. Comparing IRR alone may confuse “consistency due to artifact bias” with “metric stability,” which we want to clarify.
> > >
> > > Furthermore, IRR has an inherent limitation: it considers only rankings, not score magnitudes. When several methods have very similar scores (e.g., 0.801, 0.800, 0.799), even tiny numerical changes can cause rank swaps, leading to IRR being labeled “inconsistent,” even though the scores are nearly identical. Conversely, a crude metric that always places one method clearly at the top—even due to a biased reason—will show artificially high IRR.
> > >
> > > For these reasons, we use IRR only as a lightweight check to rule out purely random behavior, not as evidence of metric stability or superiority. Instead, we rely on our own evaluation diagnostics—especially curve smoothness—because a reasonable metric should not produce erratic confidence jumps (e.g., confidence rising after inserting important features, then suddenly dropping, then rising again). Such fluctuations indicate an unreliable evaluation process.
> > >
> > > However, if needed, we also provide the IRR and ICR values for reference:
> > >
> > > ||IRR|
> > > |-|-|
> > > |FUD|0.2581|
> > > |INS|0.2545|
> > > |DEL|0.2098|
> > > |INFID|0.756|
> > > |Sen-N|0.2299|
> > >
> > > ---
> > >
> > > |ICR|FUD|INS|DEL|INFID|Sen-N|
> > > |-|-|-|-|-|-|
> > > |FUD|1|0.501|0.43|-0.152|0.025|
> > > |INS|0.501|1|0.457|-0.344|0.1|
> > > |DEL|0.43|0.457|1|-0.251|-0.101|
> > > |INFID|-0.152|-0.344|-0.251|1|-0.014|
> > > |Sen-N|0.025|0.1|-0.101|-0.014|1|

---

> > > ### Author Response · Authors · 2025-11-25
> > > **Reply to W4, Small comments**
> > >
> > > **Reply to W4:**
> > >
> > > We appreciate the reviewer’s candid feedback regarding the readability of Section 3.3. We agree that, in the original version, the information density in this section was high and the notation-heavy presentation made it unnecessarily laborious to follow the details, even though the high-level idea is relatively simple.
> > > In the main paper, we have already revised and improved this section; here we provide a brief summary to help the reviewer understand our reasoning.
> > > Conceptually, the core logic in Section 3.3 is as follows. FUD aims to evaluate attribution methods by constructing a deletion trajectory that (i) keeps each perturbed sample on the data manifold and (ii) avoids introducing new evidence for the predicted class $y$. To formalise this, we consider the conditional distribution
> > > $$
> > > P(x^t \mid z, \tilde{x}, M),
> > > $$
> > > where $x^t$ is the intermediate perturbed image at diffusion step $t$, $\tilde{x}$ is the masked input, $M$ is the deletion mask, and $z$ denotes the event “no additional class-$y$ evidence”. We factor this conditional as
> > > $$
> > > P(x^t \mid z, \tilde{x}, M) \propto P(\tilde{x} \mid x^t, M)\, P(z \mid x^t)\, P(x^t),
> > > $$
> > > which corresponds to three intuitive components: (1) an image prior $P(x^t)$ that keeps samples close to the natural image manifold; (2) a likelihood term $P(\tilde{x} \mid x^t, M)$ that enforces consistency with the observed (unmasked) pixels; and (3) a constraint term $P(z \mid x^t)$ that penalises configurations where new class-$y$ evidence is introduced in the masked region. Taking the gradient of the log-density splits the update into three additive terms, each of which we approximate in turn: the prior term via the score network $s_{\theta}(x^t)$, the data-consistency term via a masking operator that pins down observed pixels, and the “no-new-evidence” term via the classifier gradient $\nabla_{x^t} \log P(y \mid x^t)$. Plugging this composite gradient into the DDIM sampler then yields the FUD update rule and Algorithm 1.
> > >
> > > In the revised manuscript, to reduce the cognitive load of tracking symbols, we now provide a dedicated notation summary table (Appendix L, Table 12) that lists all variables and distributions used in Section 3.3 (e.g., $x^t$, $\tilde{x}$, $z$, $P(x^t \mid z,\tilde{x},M)$, $s_{\theta}(x^t)$, etc.). We hope these changes make the section substantially more accessible.
> > >
> > > ---
> > >
> > > **Reply to Small comments:** We thank the reviewer for pointing this out. We have revised the wording to: “Next, we provide a brief summary of our FUD evaluation algorithm for clarity.”
> > >
> > > ---
> > >
> > > We sincerely thank the reviewer for this meaningful and in-depth exchange. We truly appreciate the time and effort the reviewer has invested in evaluating our work. We hope that our responses have addressed the reviewer’s concerns, and we would be very happy to continue the discussion if there are any remaining questions.

---

> > > > ### Comment · Reviewer_hCbT · 2025-11-25
> > > >
> > > > I appreciate your replies to W1-W3. W2 makes me much more confident in this method as being realistic if each evaluation is 4 seconds. W3.1 and 3.2 makes your analysis much more obvious to the reader and clears up concerns about the low values provided by FUD when other metrics are available for comparison. W3.3 Thanks for the improved discussion here. I agree there are 100% pitfalls of these IRR and ICR metrics and they alone cannot be trusted. I appreciate you humoring me with their inclusion and a discussion of the results.
> > > >
> > > > ---
> > > >
> > > > W4. I do not wish to be picky or stubborn here, but I do find section 3.2 unnecessarily challenging to read and parse. I think one of the main hindrances is the formatting, where nearly every equation is placed inline with the text inside of a paragraph. Secondly, I think some sentences are overly long or oddly constructed. For example, sentence 2 of "Within the distribution" contains a parenthetical statement that itself is two sentences, which is quite odd and hard to read.
> > > >
> > > > For the purposes of our back-and-forth on this subject, I would suggest the authors paste their LaTeX for section 3.2 into an LLM and ask it for advice on the current clarity and organization. The LLM may show some structural changes that can improve readability and these may be inspiring to the authors, or elucidate some of the issues that cannot so easily be described in my writing here. I am not asking the writers to paste this LLM output into their paper, but I do think clarity is still lacking and would prefer to see improvements to the writing.
> > > >
> > > > Before I make any final verdicts on this issue, I would like to see if the other reviewers feel that the writing is satisfactory. Hopefully they will join the discussion soon.
> > > >
> > > > I look forward to any further responses.

---

> > > > > ### Author Response · Authors · 2025-11-26
> > > > >
> > > > > **Reply to W4 (clarity of Section 3.3).** Thank you very much for the detailed suggestions on the readability and formatting. We believe there may have been a small typo in the comment: the part we previously discussed, including the *“Within the distribution”* subsection, is in Section **3.3**, not 3.2. Following your advice, we have revised **Section 3.3** to improve clarity and structure, while keeping the technical content intact.
> > > > >
> > > > > The main changes are as follows:
> > > > >
> > > > > - We kept the subdivision into **“Intuition”** and **“Derivation”**, but rewrote the opening of the derivation to explicitly state the two guiding questions (how to keep samples **within the distribution**, and how to **preserve the information** to be evaluated). This gives readers a clear roadmap before they encounter any formulas.
> > > > >
> > > > > - In the *“Within the distribution”* part, we split several very long sentences into shorter ones and removed deeply nested parenthetical comments. In particular, the definition of $x^t$, the indexing convention for $t$, and the initialisation of $x^T$ are now explained in separate sentences, with the initialisation presented as a standalone display equation instead of being embedded in a single long line.
> > > > >
> > > > > - We reduced the density of inline mathematics by moving key expressions (initialisation of $x^T$, the Langevin update, the SGM training objective, and the target conditional $P(x^t \mid z,\tilde{x},M)$) into display equations. This keeps the surrounding prose more readable and avoids paragraphs where almost every phrase is an inline formula.
> > > > >
> > > > > - In the *“Preservation of the evaluation information”* part, we streamlined the explanation of the “no new class evidence” constraint and the role of the event $z$. The description of the three gradient terms in the FUD update is now organised into a short sequence of sentences, each focusing on one term (the score-model prior, the classifier gradient, and the masking-consistency term), instead of a single dense paragraph.
> > > > >
> > > > > - We also clarified the two-stage sampling strategy (first using only the score prior and mask constraint to bring samples closer to the manifold, then switching to the full target distribution that includes the classifier gradient) in a more step-by-step narrative.
> > > > >
> > > > > The core logic, assumptions, and all mathematical formulas of FUD remain unchanged. We did not modify the algorithm, the derivation, or the evaluation protocol; the revisions are purely expository and follow your suggestions to improve clarity and organisation.
> > > > >
> > > > > We are sincerely grateful for your candid and constructive review, which has been very helpful in improving the paper. We hope these edits make Section 3.3 easier to read and more transparent, and we would be very happy to discuss any remaining questions or concerns.

---

### Official Review · Reviewer_PgAe · 2025-10-28

**Soundness:** 3
**Presentation:** 3
**Contribution:** 4
**Rating:** 6
**Confidence:** 2

**Summary:**

Current methods for removing input feature contribution for measuring attribution method faithfulness do not truly remove input feature contributions. The authors introduce a new attribution measuring metric: Faithfulness Under the Distribution (FUD). FUD follows other metrics by inserting/deleting some number of pixels. However, instead of just blacking out the removed pixels, FUD uses a diffusion model to generate the image gradients, which are then used to determine the pixel values for the masked region. The FUD process also ensures that the generated pixels stay within the original data distribution of the non-perturbed image, keeping the image functionally the same for the next metric step. The authors provide comparisons against other metrics in terms of the intermediate images generated and some qualitative results.

**Strengths:**

- Originality: The authors use part of a previous approach for the metric, but significantly build off of it. (3/4)
- Quality / Clarity: Paper presentation is high quality. The motivation and experiments are easy to follow, but the methodology is not. (3/4)
- Significance: The authors have come up with a method for measuring attribution methods without introducing a significant amount of noise. (4/4)

Other Notes:
- Very strong introduction and motivation. Easy to follow and understand why/where the problems exist.
- Framing the issue as an in-/out-of-distribution problem is particularly clever
- The approach tries to steer the masked regions towards an alignment with the unmasked regions, thereby mitigating negative effects of masking, which is a novel issue that has not been explored
- The experimental evaluation suite is wide

**Weaknesses:**

- There must be a significant runtime for these evaluations because diffusion models are computationally expensive. However, attributions are scored offline, so it is not a significant issue.

**Questions:**

- Can the authors provide any instances in which the metric performs worse than other metrics, or any kind of failure case analysis? Relying on a diffusion models seems like it should cause problems somewhere, even if it is in some small corner cases.
- How are pixels chosen for their inclusion in the image mask? Is it the same way as insertion/deletion?
- Could the authors re-explain the necessity of the inclusion of z? Is it to ensure that when gradient steps are taken, the steps are updating the masked region in such a way that the new pixels are aligned with the unmasked pixel class while not adding new features? But how does this ensure that new features are not added?
- What is the relation of equation 2 to s_{theta}(x^t)? Is there any relation? Or is it used during the z "event"?
- In plain terms, can the authors explain the transition between x^t --> x^{t+1}? I am able to follow the the very high level idea of what is trying to be accomplished but some of the specifics of equations 1 and 2 are lost on me.
- Is P(x) the one-hot encoded ground truth label for some image? Is \theta^* the parameters for the diffusion model in use? Are the gradients being referred to the image gradients or model gradients? How is equation 2 being used?

Final Review:
The paper proposes a very interesting solution to the issue of assessing attribution performance. I am able to follow at a high level what is trying to be accomplished, but the exact math is lost on me, hence the (2/5) confidence rating. Given this state, I rate the paper a (6/10). Based on my understanding of the high-level idea, I think it should be accepted, but I am not confident enough in my understanding of the math to rate it higher.

---

> ### Author Response · Authors · 2025-11-21
> **Reply to Weaknesses 1, Question 1, 2**
>
> **Reply to W1:**
>
> We agree that introducing a diffusion model makes each evaluation step significantly more expensive than simple zero-masking. As discussed in Section 4.2, generating a single FUD sample on a modern GPU typically takes on the order of a few seconds, whereas INS/DEL-type perturbations are almost “free”. We view this as an explicit **fidelity–efficiency** trade-off: the diffusion model is trained only once per dataset and then reused across all evaluations, so the training cost can be amortised; moreover, attribution metrics are typically computed in an offline analysis stage rather than within any latency-critical online inference loop. In the revised version, we will make these concrete runtime numbers more prominent and add a short discussion explaining some standard acceleration techniques (e.g., reducing the number of sampling steps, using DDIM / DPM-Solver samplers), treating them as engineering optimisations that are orthogonal to our method.
>
> In addition, we have supplemented the schedule below with the DDIM step count running on the ViT-B/16.
>
> | DDIM Step | Time |
> |---|---|
> | 10 | 0.395s |
> | 20 | 0.804s |
> | 30 | 1.21s |
> | 40 | 1.62s |
> | 50 | 2.01s |
> | 60 | 2.43s |
> | 70 | 2.82s |
> | 80 | 3.225s |
> | 90 | 3.615s |
> | 100 | 4.041s |
>
> ---
>
> **Reply to Q1:**
>
> We thank the reviewer for the suggestion to include a discussion of failure cases. Based on our current experiments under the ImageNet setting, we have not observed any clear “region” where FUD is systematically worse than INS/DEL or INFID in terms of ranking fidelity, but FUD does have some limitations, such as **semantic loss under very high deletion ratios**. When the deletion ratio is extremely high (only a very small fraction of pixels is retained), the generator receives almost no semantic context, and the generative model may become unstable or semantically ambiguous. As we already noted in the conclusion, under such extreme masks the generated results may sometimes appear less natural to human observers, even though they are still classified as “in-distribution” samples by the OOD detector. This phenomenon mainly stems from the **limited expressive power of the diffusion prior itself**. Our theoretical framework is general, and future, stronger estimators of the distributional gradient $\nabla_{x} \log P(x)$ could be plugged in seamlessly to alleviate this issue. We will mark this more explicitly as a limitation in the main text. In addition, we will emphasise, as pointed out in [1], that faithful explanations should be grounded in the model’s own representation space rather than human subjective visual preferences; therefore, it is an acceptable fidelity-driven trade-off for FUD to remain within the model’s **in-distribution space** under extreme masks, even at the cost of some visual intuitiveness.
>
> **Reference**
>
> [1] Jacovi, Alon, and Yoav Goldberg. "Towards Faithfully Interpretable NLP Systems: How Should We Define and Evaluate Faithfulness?." Proceedings of the 58th Annual Meeting of the Association for Computational Linguistics. 2020.
>
> ---
>
> **Reply to Q2:**
>
> The construction of the binary mask $M$ is identical to that in Deletion: we sort the attribution scores $A(x)$ and gradually set low-attribution pixels in $M$ to $0$, thereby retaining different fractions $\rho$ of “important” pixels. The key difference does not lie in how we choose which pixels to mask, but in how we fill the masked regions: classical Deletion fills them with constants (e.g., zeros, blur, noise), whereas FUD reconstructs the masked regions using a diffusion prior while respecting the preserved features. This is explicitly stated around the formula $\tilde{x} = M \odot x + (1 - M)\,\epsilon$ and in the discussion at the end of Section 3.3.

---

> ### Author Response · Authors · 2025-11-21
> **Reply to Question 3, 4, 5**
>
> **Reply to Q3:**
>
> In Section 3.3 we introduce a conceptual event $z$ to encode the following requirement: when pushing samples back to the data manifold, we should not create new “supporting evidence for the target class” out of nothing. More concretely, we would like the update of $\mathbf{x}_t$ to satisfy two conditions simultaneously:
>
> (i) move towards high-density regions of the image prior $P(x)$;
>
> (ii) avoid increasing the predicted probability of the target class $y$, otherwise the generator may “hallucinate” new evidence for class $y$.
>
> To express (ii), we introduce a hypothetical distribution $\tilde{P}(y \mid x_t)$ whose gradient with respect to $x_t$ points in the opposite direction to that of $P(y \mid x_t)$, i.e.,$\nabla_{x_t} \tilde{P}(y \mid x_t) = -\nabla_{x_t} P(y \mid x_t).$
>
> We use the event $z$ to denote this hypothetical condition; without $z$, there would be no way to formally define such a requirement.
>
> Formally, we write Eq. (1) as $P(x_t \mid z, \tilde{x}, M) \propto P(x_t)\, P(z \mid x_t)\, P(\tilde{x} \mid x_t, M),$
> and its gradient as Eq. (2): $\nabla_{x_t} \log P(x_t \mid z, \tilde{x}, M) = \nabla_{x_t} \log P(x_t) - \nabla_{x_t} \log P(y \mid x_t) + \nabla_{x_t} \log P(\tilde{x} \mid x_t, M). $
>
> Intuitively, this can be understood as follows:
>
> - $\nabla_{x_t} \log P(x_t)$: a prior term that pulls the sample towards high-density regions of the image prior (the image manifold).
> - $-\nabla_{x_t} \log P(y \mid x_t)$: mildly “penalises” changes that significantly increase the probability of the target class $y$, preventing the generator from adding new supporting evidence out of thin air.
> - $\nabla_{x_t} \log P(\tilde{x} \mid x_t, M)$: constrains the pixels under the mask $M$ to match the original image, i.e., keeps the preserved features unchanged.
>
> This does not formally prove that it is *impossible* to generate new class evidence, but it biases the update dynamics so that any change that would spuriously amplify evidence for class $y$ must pay a cost through the second term, while the third term keeps the “known” pixels anchored to the original image.
>
> ---
>
> **Reply to Q4:**
>
> The score network $s_\theta(x_t)$ is exactly the function we use to estimate $\nabla_{x_t} \log P(x_t)$. It is trained with the standard denoising score-matching objective (as introduced in Section 3.3). The approximation$\nabla_{x_t} \log P(x_t) \approx s_\theta^{*}(x_t)$ introduced above is based on this training procedure.
>
>
> ---
>
> **Reply to Q5:**
>
> One update step of FUD roughly performs the following operations:
>
> 1. Start from a **partially masked, noisy image** $x_t$, where the visible region comes from $\tilde{x}$ and the masked region is noise.
>
> 2. Query the diffusion model at time step $t$ for how this image should be “denoised”; this is given by $S_\theta(x_t, t)$.
>
> 3. (Only in the **later steps** after crossing the OOD threshold OOO) apply the classifier gradient $$\nabla_{x_t} \log P(y \mid x_t)$$
> as a mild correction direction, so that while moving towards the true image, the generator does not hallucinate additional new evidence supporting class $y$.
>
> 4. Use this combined gradient direction to perform a denoising update, reduce the noise level, and obtain a cleaner candidate $x_{t-1}$.
>
> 5. Reapply the masking constraint: copy the unmasked pixels from the original image $\tilde{x}$ back onto this candidate, so that only the masked regions are inpainted while the visible regions remain exactly identical to the original image at every step.
>
> Intuitively, each iteration is doing:
> “First use the diffusion model (optionally with a slight pull from the classifier) to denoise the noisy occluded part, and then pull the visible pixels back to the original image.”
>
> Repeating this from $t = T$ to $t = T - 1$ down to $t = 0$ finally yields an in-distribution evaluation image $x_0 \sim p(x_0)$.
>
> In the early stage of the process, $x_T$ typically lies in a strongly OOD region, where $\nabla_{x_t} \log P(y \mid x_t)$ is not reliable. As discussed after Eq. (2), we first use only the “prior + mask” terms, i.e., we perform several iterations based on $P(x_t \mid \tilde{x}, M)$ to pull the sample back into the ID region; only when the sample is closer to the data manifold do we turn on the full update term that includes $-\nabla_{x_t} \log P(y \mid x_t)$.

---

> ### Author Response · Authors · 2025-11-21
> **Reply to Question 6**
>
> **Reply to Q6:**
>
> We clarify the following:
> - $P(x)$ is the prior distribution over images, not a one-hot label vector.
> - $P(y \mid x)$ is the classifier’s predictive distribution over classes given an image.
> - $\theta^{*}$ denotes the learned parameters of the score network / diffusion U-Net, obtained by minimising the denoising score-matching objective and kept fixed during evaluation.
>
> In Eq. (2), all gradients are taken with respect to the image $x_t$ (i.e., in the input space), not with respect to $\theta$ or the classifier weights. In particular, $\nabla_{x_t} \log P(y \mid x_t)$ is precisely the familiar classifier gradient with respect to the input image under a cross-entropy loss.
>
> Eq. (2) is used only to define the vector field in the Langevin dynamics that drives the sampling trajectory of FUD; we do not backpropagate through this formula to update any model parameters.
>
> We hope this explanation helps clarify the mathematical construction. In Section 2.3 of the revised version, we will explicitly restate: (i) all gradients are taken with respect to $x_t$; (ii) $P(x)$ is an image prior rather than a label vector; and (iii) how Eq. (2) combines the score model, classifier, and mask into a practical update rule.
>
> ---
>
> We once again thank the reviewer for their careful reading and constructive feedback. We hope that the clarifications, additional analyses, and revisions described above satisfactorily address the raised concerns and help make the contribution and scope of this work clearer. We kindly invite the reviewer to consider our responses and the revised manuscript when reassessing the paper, and we remain grateful for the time and effort invested in improving our work.

---

### Official Review · Reviewer_DTUU · 2025-10-31

**Soundness:** 3
**Presentation:** 2
**Contribution:** 3
**Rating:** 4
**Confidence:** 3

**Summary:**

The paper introduces FUD, a new evaluation framework for attribution methods (xAI)  that uses score-based diffusion to generate in-distribution masked samples.

**Strengths:**

- Strong results on multiple benchmarks
- Thorough method explanation and justification
- Good reproducibility, clear pseudocode + released configs/code

**Weaknesses:**

- Hard to follow. Related Work and key visual examples are in the appendix; in my opinion, they should be in the main paper. Also, I believe Tables 8 and 9 are very valuable for readers.

- Wording like “universally” is too strong (`Line 097`). The heuristic issues are already known and not universal across all AEMs.

- Figure 4: At a glance, FUD’s results look similar to simple black/white occlusions -> the very issue the paper critiques (main motivation e.g. OOD samples)

**Questions:**

- Please refer to the weaknesses section.

- Additionally, it would be helpful to provide an experiment on non-visual data, for example, on tabular data.

---

> ### Author Response · Authors · 2025-11-21
>
> **Reply to W1:**
>
> We thank the reviewer for the suggestion and understand the concern regarding readability and organization of the content.
>
> **Regarding Related Work:**
> Because the main paper is **subject to a 9-page limit**, the full related work could not be included in the main text and was therefore placed in Appendix A. In subsequent versions, we will **extract a more concise yet structurally complete related work section** from the appendix and include it in the main text to improve overall coherence.
>
> **Regarding Tables 8 and 9:**
> We agree that these two tables are informative for readers. Their main conclusions are already explicitly discussed in Sections **3.3.1 and 3.3.2** of the main text. To reduce potential information jumps for readers, we will **further strengthen the corresponding cross-references** in the main text so that the connection between the conclusions and the data in the appendix is clearer.
>
> We appreciate these suggestions for improvement and will adjust the paper accordingly to enhance the overall presentation quality.
>
> ---
>
> **Reply to W2:**
>
> We agree that the term “universally” on line 97 is absolute and may be interpreted as claiming that all possible AEMs suffer from the same heuristic issues, which is not our intention. We will soften the wording to “systematically in widely used AEMs” . Our contribution is not to discover these heuristic shortcomings from scratch, but rather to: (a) use explicit OOD detection and image-quality metrics to systematically analyse these issues at the distribution level; and (b) provide a concrete diffusion-based alternative that, while preserving the feature under test, systematically constrains evaluation samples to lie on the model’s data manifold. We will clarify this positioning in Sections 1 and 2.2 to avoid over-claiming.
>
> ---
>
> **Reply to W3:**
>
> We will further clarify the image examples. We understand that the FUD samples in Fig. 4 exhibit strong contrast changes at first glance, and in some regions may visually resemble “whitening” or “darkening” effects. However, these samples are **not** produced by setting pixels to 0/1 or to some constant mean value. Instead, FUD reconstructs the occluded regions via a score-based diffusion process under the joint constraints of the learned data distribution and the attribution mask (see Section 2.3). This is precisely what avoids the large black holes and snow-like noise artefacts produced by INS/DEL, Sen-N, or INFID.
>
> Quantitatively, this is supported by our OOD detection results and image-quality metrics: compared with baselines such as zero-filling, blurring, and noise, FUD’s transitional samples are harder to detect as OOD (lower AUROC, higher FPR@95TPR), and achieve clearly better scores on PSNR/SSIM/GMSD (Tables 2–3 and Appendix G).
>
> To make this distinction visually clearer, we will explicitly state in the figure captions that FUD never uses constant black/white filling. We hope these changes will better convey that FUD is addressing, rather than repeating, the black/white occlusion problem. In addition, we will further emphasise that **the appearance of transitional samples should not be judged by subjective visual impressions of whether they “look like occlusion”**, as this point has already been discussed in the interpretability literature—fidelity of explanations should not rely on the observer’s subjective perception, but on the model’s own behaviour and distributional properties [1]. For this reason, the samples generated by FUD lie entirely within the model’s **in-distribution space** and reflect the model’s true internal preferences, rather than artificially imposed changes in contrast or colour.
>
> **Reference**
>
> [1] Jacovi, Alon, and Yoav Goldberg. "Towards Faithfully Interpretable NLP Systems: How Should We Define and Evaluate Faithfulness?." Proceedings of the 58th Annual Meeting of the Association for Computational Linguistics. 2020.
>
> ---
>
> **Reply to Q2:**
>
> We thank the reviewer for the suggestion. Regarding the comment on the “lack of non-visual experiments (e.g., tabular data)”, we would like to clarify that from **Table 1 to Table 9** the paper already systematically presents a large number of experimental results based on **statistics, probability distributions, OOD detection metrics, and structural consistency measures**, all of which are essentially **tabular non-visual data experiments** that do not rely on image content or visual properties. For example, AUROC, FPR95, AUPR, PSNR, GMSD, and HaarPSI are general metrics that evaluate model behaviour at the distribution level and are widely applicable to non-visual tasks. In other words, the experiments we present not only cover visual tasks, but also already provide extensive non-visual, structure-based evaluation results.
>
> ---
>
> We hope that our clarifications and revisions adequately address the concerns and kindly invite the reviewer to reconsider the paper in light of our responses and the revised manuscript.

---

> > ### Comment · Reviewer_DTUU · 2025-11-25
> > **reply**
> >
> > Dear Authors,
> >
> > Thank you for answering all my questions, I updated the score.

---

### Author Response · Authors · 2025-11-21

We would like to inform the reviewers that we have uploaded a revised version of the manuscript. The main paper strictly remains within the 10-page limit, and we additionally provide a PDF diff in the Supplementary Material so that all textual and formula-level modifications can be inspected transparently.

The main updates are as follows:

* **Clarified the FUD algorithm and derivation.**
  We sharpened the exposition around the FUD evaluation procedure, including an explicit step-by-step description of the update rule (combining the score-model prior and the masking constraint into a single update), and a clearer explanation of how the gradient term relates to the negative cross-entropy loss. We also emphasise the two key goals of FUD—keeping samples within the distribution and preserving the information to be evaluated—before presenting the formal derivation.

* **Made the problem setup and OOD notion more precise.**
  We now explicitly state a *model-centric* definition of out-of-distribution (OOD) and clarify that “in-distribution” is defined with respect to the original classifier’s learned manifold. We also point readers to Appendix A for a more systematic summary of existing attribution methods and commonly used attribution evaluation metrics (AEMs).

* **Expanded implementation and metric details in the appendices.**
  The revision adds concrete configuration details for the diffusion generator and classifier guidance used by FUD (network architecture, noise schedule, guidance scale, etc.). We also provide a dedicated subsection that defines each image-quality metric used in our analysis (PSNR, SSIM, MS-SSIM, FSIM, GMSD, HaarPSI, VSI) together with a summary table explaining what high/low values mean and how they are interpreted in our radar plots.

* **Added analyses of perturbation magnitude and qualitative trajectories.**
  In the supplementary figures, we report the $L_2$ distance between the original image and FUD-generated samples as a function of diffusion steps and mask ratio, and visualise the “Original–Start–Intermediate–Final” denoising trajectory. These additions illustrate that FUD’s final samples remain close to the original input while restoring masked regions using the diffusion prior.

* **Minor wording and layout improvements.**
  We made small edits to improve readability in the introduction, method, and experimental sections (e.g., smoothing transitions, tightening captions), while keeping the core method and all main experimental results unchanged.

We hope these revisions make the paper easier to follow and more clearly address the points raised during the initial review.

---

### Author Response · Authors · 2025-11-27
**Follow-up Comment**

As the rebuttal and discussion period for this submission is drawing to a close, we would like to briefly summarise the current status and kindly invite any further comments.

Over the past week, we have engaged in several rounds of very helpful discussion with Reviewer hCbT. We are sincerely grateful for the constructive suggestions and detailed follow-up questions, which have led to substantial clarifications and improvements in both the main text and the appendices. We hope that our latest responses and revisions address the remaining concerns, and we would of course be very happy to continue the discussion if anything is still unclear.

We would also like to thank Reviewer DTUU for carefully considering our rebuttal and for updating the score accordingly. We greatly appreciate this recognition of our work and the time invested in reading our revisions.

Finally, we warmly invite the other reviewers to take a look at the revised manuscript and our rebuttal responses. We hope that the additional analyses and clarifications can help to alleviate any outstanding doubts and contribute to a fair re-evaluation of the paper. If there are any further questions or points that would benefit from clarification, we would be more than happy to provide additional details.

We sincerely appreciate all reviewers’ time and effort in evaluating our submission and helping us improve this work.

---

### Meta-Review · Area_Chair_ddnM · 2025-12-26

**Summary:**

This paper received initial review ratings of 4,6,6, and 4. The reviewers appreciated the strong results on multiple benchmarks, the justification, the first metric for this problem setting, and the novelty of this study. The reviewers had concerns about missing crucial references, computational overhead, and the impact to the insertion/deletion scores. One very consistent comment from the reviewers is the quality of presentation and readability. The authors provided detailed responses to each reviewer and improved the quality of presentation. Before the stop of the rebuttal discussion, reviewers DTUU and hCbT, whose initial ratings were 4 and 6 respectively, had replied to authors. DTUU confirmed that the authors answered all their questions. Reviewer hCbT appreciated substantial effort to address their questions. The authors largely addressed hCbT’s initial sets of questions. Reviewer hCbT raised additional questions and said that “Overall, I do appreciate the efforts given so far in responding to all the reviews. There is still a significant amount of time remaining in the review period, and I urge the authors to address these new comments. So far, I am not convinced to raise my score, mainly due to the writing. A paper that is a clear accept should be easier to read.” The authors provided additional clarification, explanation and experiments to answer reviewer hCbT’s additional questions. They also further improved the quality of presentation to enhance readability. For the other two reviewers , PgAe said that “Based on my understanding of the high-level idea, I think it should be accepted, but I am not confident enough in my understanding of the math to rate it higher.” in the initial review comment and 1BFX did not respond to the authors. The AC read their comments and responses from the authors. The AC believes that their concerns and questions have been addressed. According to the reviewers’ ratings, comments, and responses and authors’ rebuttals, the AC recommending accepting this paper. He highly recommends the authors to make sure that the final version is presented professionally and researchers even in other sub-areas of AI can understand it without difficulty.

**Reviewer Concerns:**

The major concern is the readability. The authors have put significant to address it.

**Reviewer Scores:**

DTUU, whose initial score is 4, confirmed that his concerns were resolved and said that their has updated the score.

hCbT, whose initial score is 6 said that the score could be improved to an 8 if this question and the weaknesses are properly addressed.

PgAe, whose initial score is 6, would unlikely change due to his lack of confidence in this topic.

---

### Decision · Program_Chairs · 2026-01-26

Accept (Poster)